
# Diapycnal mixing across the photic zone of the NE-Atlantic

by Hans van Haren*, Corina P.D. Brussaard, Loes J. A. Gerringa, Mathijs H. van Manen, Rob Middag, Ruud Groenewegen

Royal Netherlands Institute for Sea Research (NIOZ), P.O. Box 59, 1790  AB Den Burg, the Netherlands.
*e-mail: hans.van.haren@nioz.nl

**Abstract.** Variable physical conditions such as vertical turbulent exchange, internal wave and
mesoscale eddy action, affect the availability of light and nutrients for phytoplankton
(unicellular algae) growth. It is hypothesized that changes in ocean temperature may affect
ocean vertical density stratification, which may hamper vertical exchange. In order to quantify
variations in physical conditions in the Northeast Atlantic Ocean, we sampled a latitudinal
transect along 17±5°W between 30 and 63°N in summer. A shipborne Conductivity-
Temperature-Depth CTD-instrumented package was used with a custom-made modification of
the pump-inlet to minimize detrimental effects of ship motions on its data. Thorpe-scale
analysis was used to establish turbulence values for the upper 500 m from 3 to 6 profiles
obtained in a short CTD-yoyo, 3 to 5 h after local sunrise. From south to north, average
temperature decreased together with stratification while turbulence values weakly increased or
remained constant. Vertical turbulent nutrient fluxes did not vary significantly with
stratification and latitude. This apparent lack of correspondence between turbulent mixing and
temperature is likely due to internal waves breaking (increased stratification can support more
internal waves), acting as a potential feed-back mechanism. As this feed-back mechanism
mediates potential physical environment changes in temperature, global surface ocean warming
may not affect the vertical nutrient fluxes to a large degree. We urge modelers to test this
deduction as it could imply that the future summer phytoplankton productivity in stratified
oligotrophic waters would experience little alterations in nutrient input from deeper waters.


## 1 Introduction

The physical environment is important for ocean life, including variations therein. For example, the sun stores heat in the ocean with a stable vertical density stratification as result. Generally, stratification hampers vertical turbulent exchange because of the required work against (reduced) gravity before turbulence can take effect. It thus hampers a supply of nutrients via a turbulent flux from deeper waters to the photic zone. However, stratification supports internal waves, which (i) may move near-floating particles like phytoplankton (unicellular algae) up- and down towards and away from the surface, and (ii) may induce enhanced turbulence via vertical current differences (shear) resulting in internal waves breaking (Denman and Gargett, 1983). Such changes in the physical environment are expected to affect the availability of phytoplankton growth factors such as light and nutrients.

Climate models predict that global warming will reduce vertical mixing in the oceans (e.g., Sarmiento et al., 2004). Mathematical models on system stability suggest that reduced mixing may generate chaos behaviour in phytoplankton production, thereby enhancing variability in carbon export into the ocean interior (Huisman et al., 2006). However, none of these models include potential feed-back systems like internal wave action or mesoscale eddy activity. From observations in the relatively shallow North Sea it is known that the strong seasonal temperature stratification is marginally stable, as it supports internal waves and shear to such extent that sufficient nutrients are replenished from below to sustain the late-summer phytoplankton bloom in the euphotic zone that became depleted of nutrients after the spring bloom (van Haren et al., 1999). This challenges the current paradigm in climate models.

In this paper, the objective is to resolve the effect of vertical stratification and turbulent mixing on nutrient supply to the euphotic zone of the open ocean. For this purpose, upper-500-m-ocean shipborne Conductivity-Temperature-Depth CTD-observations were made in association with those on dissolved inorganic nutrients during a survey along a transect in the NE-Atlantic Ocean from mid-(30°) to high--(63°) latitudes in summer. Throughout the survey, meteorological and sea-state conditions were favourable for adequate sampling and wind

speeds varied little between 5 and 10 m s$^{-1}$, independent of locations. All CTD-observations
were made far from lateral, continental boundaries and at least 1000 m vertically away from
bottom topography (i.e. far from internal-tide sources). The NE-Atlantic is characterized by
abundant (sub-)mesoscale eddies especially in the upper ocean (Charria et al., 2017) that
influence local plankton communities (Hernández- Hernández et al., 2020). The area also
shows continuous abundant internal wave activity away from topographic sources and sinks,
with the semidiurnal tide as a main source from below and atmospherically induced inertial
motions from above (e.g., van Haren, 2005; 2007). However, the sampled upper 500-m zone
transect is not known to demonstrate outstanding internal wave source variations. Previous
observations (van Haren, 2005) and Hibiya et al. (2007) have shown that a diurnal critical
latitude enhancement of near-inertial internal waves due to subharmonic instability only occurs
sharply between 25 and 30°N. The present observations are all made poleward of this range.
Likewise, the Henyey et al. (1986) model on latitudinal variation of internal wave energy and
turbulent mixing (Gregg et al., 2003) predicts changes by a factor of maximum 1.8 between
30° and 63°, but this value is relatively small compared with errors, typically a factor of 2 to 3,
in turbulence dissipation rate observations. Likewise, from the equal summertime
meteorological conditions little variation is expected in the generation of upper ocean near-
inertial internal waves. Naturally, other processes like interaction between internal waves and
mesoscale phenomena may be important locally, but these are expected to occur in a similar
fashion across the sampled ocean far away from boundaries. Thus, the sampled dataset is
considered adequate for a discussion on the variability of turbulence, stratification and vertical
turbulent nutrient fluxes with latitude.
The present research complements research based on photic zone (upper 100 m)
observations obtained along the same transect using a slowly descending turbulence
microstructure profiler next to CTD-sampling eight years earlier (Jurado et al., 2012). Their
data demonstrated a negligibly weak increase in turbulence values with significant decreases in
stratification going north. However, no nutrient data were presented and no turbulent nutrient
fluxes could be computed. In another summertime study (Mojica et al., 2016), macro-nutrient
concentrations indicated oligotrophic conditions along the same latitudinal transect but the
vertical gradients for the upper 200 m showed an increase from south to north. The present
observations go deeper to 500 m, also across the non-seasonal more permanent stratification.
Moreover, coinciding measurements were made of the distributions of macro-nutrients and
dissolved iron. This allows vertical turbulent nutrient fluxes to be computed. It leads to a
hypothesis concerning a physical feed-back mechanism that may control changes in
stratification.

**2 Materials and Methods**
Between 22 July and 16 August 2017, observations were made from the R/V Pelagia in the
Northeast Atlantic Ocean at stations along a transect from Iceland, starting around 60°N, to the
Canary Islands, ending at 30°N, (Fig. 1). The transect was roughly in meridional direction, with
stations along $17\pm5$°W, all in the same time zone (UTC-1 h = local time LT). Full water-depth
Rosette bottle water sampling was performed at most stations.
Samples for dissolved inorganic macro-nutrients were filtered through 0.2 μm Acrodisc
filter and stored frozen in a high-density polyethylene pony-vial (nitrate, nitrite and phosphate)
or at 4°C (silicate) until analysis. Nutrients were analysed under temperature controlled
conditions using a QuAAtro Gas Segmented Continuous Flow Analyser. All measurements
were calibrated with standards diluted in low nutrient seawater in the salinity range of the
stations to ensure that analysis remained within the same ionic strength. Phosphate ($PO_4$),
nitrate plus nitrite ($NO_x$), were measured according to Murphy and Riley (1962) and Grasshoff
et al. (1983), respectively. Silicate was analysed using the procedure of Strickland and Parsons

(1968).

Absolute and relative precision were regularly determined for reasonably high
concentrations in an in-house standard. For phosphate, the standard deviation was 0.028 μM
(N = 30) for a concentration of 0.9 μM; Hence the relative precision was 3.1%. For nitrate, the
values were 0.14 µM (N = 30) for a concentration of 14.0 µM, so that the relative precision was
1.0%. For silicate, the values were 0.09 µM (N = 15) for a concentration of 21.0 µM, so that
the relative precision was 0.4%. The detection limits were 0.007, 0.012 and 0.008 µM, for
phosphate, nitrate and silicate, respectively.
For dissolved iron samples, the ultraclean "Pristine" sampling system for trace metals was
used (Rijkenberg et al., 2015). All bottles used for storage of reagents and samples were cleaned
according to an intensive three step cleaning protocol described by Middag et al. (2009).
Dissolved iron concentrations were measured shipboard using a Flow Injection—
Chemiluminescence method with preconcentration on iminodiaceticacid resin as described by
De Baar et al. (2008) and modified by Klunder et al. (2011). In order to validate the accuracy
of the system, standard reference seawater (SAFe) was measured regularly in triplicate
(Johnson et al. 1997).
At 19 out of 32 stations a yoyo consisting of 3 to 6 casts, totaling 88 casts, of electronic
CTD profiles was done to monitor the temperature-salinity variability and to establish turbulent
mixing values from 5 to 500 m below the ocean surface. For the yoyos a separate CTD was
used from the CTD -- ultraclean sampling system. The yoyo casts were made consecutively
and took between 1 and 2 hours per station. They were mostly obtained in the morning: at ten
stations between 6 and 8 LT, at eight stations between 8 and 10 LT, and at one station in the
afternoon, around noon. As the observations were made in summer, the latitudinal difference
in sunrise was 1.5 h between the northernmost (earlier sunrise) and southernmost stations. This
difference is taken into account and sampling times are referenced to time after local sunrise. It
is assumed that the stations sampled just after sunrise reflect the upper ocean conditions of (late-
) nighttime cooling convection so that vertical near-homogeneity was at a maximum, and near-
surface stratification at a minimum, while the late morning and afternoon stations reflected
daytime stratifying near-surface conditions due to the stabilizing solar insolation.

**2.1 Instrumentation and modification**

Calibrated SeaBird 911plus CTDs were used. The CTD data were sampled at a rate of 24

Hz, whilst lowering the instrumental package at an average speed of 0.9 m s$^{-1}$. The yoyo CTD
data were processed using the standard procedures incorporated in the SBE-software, including
corrections for cell thermal mass (Lueck, 1990) using the parameter setting of Mensah et al.
(2009), sensor time-alignment and vertical bin-averaging over 0.33 m. All other analyses were
performed using Conservative Temperature ($\Theta$), Absolute Salinity $S_A$ and potential density
anomalies $\sigma_\theta$, with 1000 kg m$^{-3}$ subtracted from total density and referenced to the surface for
pressure corrections as vertical profiles were only analyzed shallower than 600 m, using the
Gibbs SeaWater-software (IOC, SCOR, IAPSO, 2010).

Observations were made with the yoyo CTD upright rather than horizontal in a lead-

weighted frame without water samplers to minimize artificial turbulent overturning. Variable
speeds of the flow passing the temperature and conductivity sensors will cause artificial
temperature and thus apparent turbulent overturning, noticeable in near-homogeneous waters
such as found near the surface during nighttime convection. To eliminate variable flow speeds,
a custom-made assembly with pump in- and outlet tubes and tube-ends of exactly the same
diameter was mounted to the CTD as described in van Haren and Laan (2016). This reduces
frictional temperature effects of typically ±0.5 mK due to fluctuations in pump speed of ±0.5
m s$^{-1}$ when standard SBE-tubing is used (Appendix A1). The effective removal of the artificial
temperature effects using the custom-made assembly is demonstrated in Fig. 2, in which surface
wave action via ship motion is visible in the CTD-pressure record, but not in its temperature
variations record. For example, at station 32 the CTD was lowered in moderate sea state
conditions with surface waves of maximum 2 m crest-trough. The surface waves are recorded
by pressure variations as a result of ship motions (Fig. 2a). In the upper 35 m near the surface,
the waters were partially unstable and partially near-homogeneous, with temperature variations
well within ±0.5 mK and high-frequency variations O(0.1) mK (Fig. 2b). The $\Delta$T-variations
did not vary with the surface wave periodicity of about 10 s. No correlation was found between
data in Fig. 2b and Fig. 2a. This effective removal of ship motion in CTD-temperature data is
confirmed for the entire 500 m depth-range in average spectral information (Fig. 2c-e). In the
power spectra, the pressure gradient dp/dt ~ CTD-velocity shows a clear peak around 0.1 cps,
short for cycles per s, which correspond to a period of 10 s. Such a peak is absent in both spectra
of temperature T and density anomaly referenced to the surface $\sigma_\theta$. The correlation between
dp/dt and T is not significantly different from zero (Fig. 2d,e). With conventional tubing and
tube-ends, the surface wave variations would show in such $\Delta$T-graph (van Haren and Laan,
2016). Without the effects of ship motions, considerably less corrections need to be applied for
turbulence calculations (see below).

**2.2 Ocean turbulence calculation**
Turbulence is quantified using the analysis method by Thorpe (1977) on potential density
inversions of less dense water below a layer of denser water in a vertical (z) profile. Such
inversions are interpreted as turbulent overturns of mechanical energy mixing. Vertical
turbulent kinetic energy dissipation rate ($\varepsilon$) is a measure of the amount of kinetic energy put in
a system for turbulent mixing. It is proportional to the magnitude of turbulent diapycnal flux
(of potential density) $|K_z d\sigma_\theta/dz|$. In practice it is determined by calculating overturning scales
with magnitude $|d|$, just like turbulent eddy diffusivity ($K_z$). The vertical potential density
stratification is indicated by $d\sigma_\theta/dz$. The turbulent overturning scales are obtained after
reordering the measured profile $\sigma_\theta(z)$, which may contain inversions, into a stable monotonic
profile $\sigma_\theta(z_s)$ without inversions (Thorpe, 1977). After comparing raw and reordered profiles,
displacements $d = \min(|z-z_s|) \cdot \mathrm{sgn}(z-z_s)$ are calculated that generate the stable profile. Then,
using root-mean-square displacement value $L_T = \mathrm{rms}(d)$ computed over certain vertical scales
(see below),

$$\varepsilon = 0.64 L_T^2 N^3 \qquad [\mathrm{m^2 s^{-3}}], \tag{1}$$

where $N = \{-g/\rho(d\sigma_\theta(z_s)/dz)\}^{1/2}$ denotes the buoyancy frequency (~ square-root of stratification
as is clear from the equation) computed from the reordered profile. Here, g is the acceleration
of gravity and $\rho = 1027$ kg m$^{-3}$ denotes the reference density. We like to note, following
previous warnings by, e.g., Gill (1982) and King et al. (2012), that our definition of N is a
practical one, which should not be used for data from deeper waters. For deeper waters, density
should be referenced to a local pressure reference level, which effectively implies the use of the
exact definition for buoyancy frequency as formulated, e.g., by Gill (1982): $\{-g/\rho(d\rho/dz +$
$g\rho/c_s^2\}^{1/2}$, where $c_s$ is the speed of sound reflecting pressure-compressibility effects. Our
'surface waters' N computed over reordered profiles only negligibly deviates from above exact
N and corresponds with N computed from raw profiles over a typical vertical length-scale of
$\Delta z = 100$ m. This $\Delta z$ represents the scales of large internal waves that are supported by the
density stratification and of the largest turbulent overturns.

The numerical constant of 0.64 in (1) follows from empirically relating the overturning scale

magnitude with the Ozmidov scale $L_O$ of largest possible turbulent overturn in a stratified flow:
$(L_O/L_T) = 0.8$ (Dillon, 1982), a mean coefficient value from many realizations. Using $K_z = \Gamma \varepsilon N^-$
$^2$ and a mean mixing efficiency coefficient of $\Gamma = 0.2$ for the conversion of kinetic into potential
energy for ocean observations that are suitably averaged over all relevant turbulent overturning
scales of the mix of shear-, current differences, and convective, buoyancy driven, turbulent
overturning in large Reynolds number flow conditions (e.g., Osborn, 1980; Oakey, 1982;
Ferron et al., 1998; Gregg et al., 2018), we find,

$K_z = 0.128 L_T^2 N$       $[\text{m}^2\text{s}^{-1}]$.               (2)

This parametrization is also valid for the upper ocean, as has been shown extensively by Oakey
(1982) and recently confirmed by Gregg et al. (2018). The inference is that the upper ocean
may be weakly stratified at times, but stratification and turbulence vary considerably with time
and space. Sufficient averaging collapses coefficients to the mean values given above. This is
confirmed in recent numerical modeling by Portwood et al. (2019).

As $K_z$ is a mechanical turbulence coefficient it is not property-dependent like a molecular

diffusion coefficient that is about 100-fold different for temperature compared to salinity. $K_z$ is
thus the same for all turbulent transport calculations no matter what gradient of what property.
For example, the vertical downgradient turbulent flux of dissolved iron transporting from iron-
rich deeper waters upwards into the euphotic zone is computed as $-K_z d(DFe)/dz$.

According to Thorpe (1977), results from (1) and (2) are only useful after averaging over

the size of a turbulent overturn instead of using single displacements. Here, rms-displacement
values $L_T$ are not determined over individual overturns, as in Dillon (1982), but over 7 m
vertical intervals (equivalent to about 200 raw data samples) that just exceed average $L_O$. This
avoids the complex distinction of smaller overturns in larger ones and allows the use of a single
length scale of averaging. As a criterion for determining overturns we only used those data of
which the absolute value of difference with the local reordered value exceeds a threshold of
$7 \times 10^{-5}$ kg m$^{-3}$, which comes from standard deviations of the potential density profiles in near-
homogeneous layers over 1-m intervals and which corresponds to noise-variational amplitudes
of $1.4 \times 10^{-4}$ kg m$^{-3}$ in raw data (e.g., Galbraith and Kelley, 1996; Stansfield et al., 2001; Gargett
and Garner, 2008). Vertically averaged turbulence values, short for averaged $\varepsilon$- and $K_z$-values
from (1) and (2), can be calculated to within an error of a factor of two to three, approximately.
As will be demonstrated below, this is considerably less spread in values than the natural
turbulence values variability over typically four orders of magnitude at a given position and
depth in the ocean (e.g., Gregg, 1989).

**3 Results**
**3.1 Physical parameters**

An early morning vertical profile of density anomaly in the upper 500 m at a northern

station (Fig. 3a) is characterized by a near-homogeneous layer of about 15 to 40 m, which is
above a layer of relatively strong stratification and a smooth moderate stratification deeper
below. In the near-homogeneous upper layer, in this example z > -30 m, relatively large
turbulent overturn displacements can be found of d = ±20 m (Fig. 3b): so called large density
inversions. In this paper we conventionally define 'mixed layer depth' as the depth at which the
temperature difference with respect to the surface is 0.5°C (Jurado et al., 2012). We note that
this actually more represents the 'mixing layer depth' and the reordered profile shows non-zero
stratification. If the mixed-layer-depth definition would have been applying a temperature
difference of, e.g., 0.001°C on the reordered profile, its value would average about 5 m, much
less than using the present and more common, conventional definition applying a temperature
difference of 0.5°C. We thus present turbulence results for this commonly defined 'mixed layer'
with caution, whilst observing their consistency with the results from deeper down, as presented
below. For $-200 < z < -30$ m, large turbulent overturns are few and far between. Turbulence
dissipation rate (Fig. 3c) and eddy diffusivity (Fig. 3d) are characterized by relatively small
displacement sizes of less than 5 m. For $z < -200$ m, displacement values weakly increase with
depth, together with stratification ($\sim N^2$; Fig. 3e). Between $-30 < z < 0$ m, turbulence dissipation
rate values between our minimum detectable level $10^{-11}$ and $>10^{-8}$ m$^2$ s$^{-3}$ are similar to those
found by others, using microstructure profilers (e.g., Oakey, 1982; Gregg, 1989), lowered
acoustic Doppler current profiler or CTD-Thorpe scale analysis (e.g., Ferron et al., 1998; Walter
et al., 2005; Kunze et al., 2006). Here, eddy diffusivities are found between our minimum
detectable $2\times10^{-5}$ and $3\times10^{-3}$ m$^2$ s$^{-1}$ and these values compare with previous near-surface results
(Denman and Gargett, 1983). The relatively small $|d| < 5$ m displacements (Fig. 3b) are genuine
turbulent overturns, and they resemble 'Rankine vortices', a common model of cyclones (van
Haren and Gostiaux, 2014), as may be best visible in this example in the large turbulent overturn
near the surface. The occasional erratic appearance in individual profiles, sometimes still visible
in the ten-profile means, reflects smaller overturns in larger ones.

A mid-morning profile at a southern station shows different characteristics (Fig. 4),

although 500 m vertically averaged turbulence values are similar to within 10% of those of the
northern station. This 10% variation is well within the error bounds of about a factor of two. At
this southern station, the near-surface layer is stably stratifying (Fig. 4a) and displays few
overturning displacements (Fig. 4b), while the interior demonstrates rarer but occasional
intense turbulent overturning (at $z = -160$ m in Fig. 4), presumably due to internal wave
breaking. At greater depths, stratification ($\sim N^2$; Fig. 4e) weakly decreases, together with $\varepsilon$ (Fig.
4c) and $K_z$ (Fig. 4d).
Latitudinal overviews are given in Fig. 5 for: Average values over the upper $z > -15$ m,
which covers the diurnal mainly convective turbulent mixing range from the surface and under
the cautionary note that these waters are weakly, but measurably stratified, average values
between $-100 < z < -25$ m, which covers the seasonal strong stratification, and average values
between $-500 < z < -100$ m, which covers the more permanent moderate stratification. Noting
that all panels have a vertical axis representing a logarithmic scale, variations over nearly four
orders of magnitude in turbulence dissipation rate (Fig. 5a) and eddy diffusivity (Fig. 5b) are
observed between casts at the same station. This variation in magnitude is typically found in
near-surface open-ocean turbulence microstructure profiles (e.g., Oakey, 1982). Still,
considerable variability over about two orders of magnitude is observed between the averages
from the different stations. This variation in station- and vertical averages far exceeds the
instrumental error bounds of a factor of two (0.3 on a log-scale), and thus reveals local
variability. The turbulence processes occur 'intermittently'.
The observed variability over two orders of magnitude between yoyo-casts at a single
station may be due to active convective overturning during early morning in the near-
homogeneous upper layer, or due to internal wave breaking and sub-mesoscale variability
deeper down. Despite the large variability at stations, trends are visible between stations in the
upper 100 m over the 33° latitudinal range going poleward: Buoyancy frequency ($\sim$ square root
of stratification) steadily decreases significantly (p-value < 0.05) given the spread of values at
given stations, with the notion that near-surface ($-15 < z < 0$ m) values show the same latitudinal
trend as deeper-down-values across a larger spread of values, while turbulence values vary
insignificantly with latitude as they remain the same or weakly increase by about half an order
of magnitude (about a factor of 3). At a given depth range, turbulence dissipation rates roughly
follow a log-normal distribution with standard deviations well exceeding half an order of
magnitude. The comparison of latitudinal variations with the (log-normal) distribution is
declared insignificant with $p > 0.05$ when the mean values are found within 2 standard
deviations (see Appendix A2). This is not only performed for turbulence dissipation rate, but
also for other quantities. The trends suggest only marginally larger turbulence going poleward,
which is possibly due to larger cooling from above and larger internal wave breaking deeper
down. It is noted that the results are somewhat biased by the sampling scheme, which changed
from 3 to 4 h after sunrise sampling at high latitudes to 4 to 5 h after sunrise sampling at lower
latitudes, see the sampling hours after local sunrise in (Fig. 5d). Its effect is difficult to quantify,
but should not show up in turbulence values from deeper down ($-500 < z < -100$ m).

Between $-500 < z < -100$ m, no clear significant trend with latitude is visible in the

turbulence values (Fig. 5a,b), although $[K_z]$ weakly increases with increasing latitude at all
levels between $-500 < z < 0$ m, while buoyancy frequency significantly decreases (Fig. 5c). The
data from well-stratified waters deeper down thus show the same latitudinal trend as the
observations from the near-surface layers, even though the latter are less well determined
because of the weak stratification. Our turbulence values from CTD-data also confirm previous
results by Jurado et al. (2012) who made microstructure profiler observations from the upper $z$
$> -100$ m along the same transect. Their results showed turbulence values remain unchanged
over 30º latitude or increase by at most one order of magnitude, depending on depth level. Their
'mixed' layer ($z >\sim -25$ m) turbulence values are similar to our $z > -15$ m values and half to one
order of magnitude larger than the present deeper observations. The slight discrepancy in values
averaged over $z > -25$ m may point at either i) a low bias due to a too strict criterion of accepting
density variations for reordering applied here, or ii) a high bias of the ~10-m largest overturns
having similar velocity scales (of about 0.05 m s$^{-1}$) as their 0.1 m s$^{-1}$ slowly descending SCAMP
microstructure profiler. At greater depths, $-500 < z < -100$ m, it is seen in the present
observations that the spread in turbulence values over four orders of magnitude at a particular
station is also large. This spread in values suggests that dominant turbulence processes show
similar intermittency in weakly (at high-latitudes $N \approx 10^{-2.5}$ s$^{-1}$) and moderately (at mid-latitudes
$N \approx 10^{-2.2}$ s$^{-1}$) stratified waters, respectively, for the given resolution of the instrumentation.

Mean values of N are larger by half an order of magnitude in the seasonal pycnocline (found

in the range $-100 < z < -25$ m) than those near the surface and in the more permanent
stratification below (Fig. 5). Such local vertical variations in N have the same range of variation
as observed horizontally across latitudes [30, 63]° per depth level.

**3.2 Nutrient distributions and fluxes**

Vertical profiles of macro-nutrients generally resemble those of density anomaly in the

upper $z > -500$ m (Fig. 6). In the south, low macro-nutrient values are generally distributed over
a somewhat larger near-surface mixed layer. The mixed layer depth, at which temperature
differed by 0.5°C from the surface (Jurado et al., 2012), varies between about 20 and 30 m on
the southern end of the transect and weakly becomes shallower with latitude (Fig. 7a). This
weak trend may be expected from the summertime wind conditions that also barely vary with
latitude (Fig. 7b,c). In contrast, the euphotic zone, defined as the depth of the 0.1% irradiance
penetration level (Mojica et al., 2015), demonstrates a clear latitudinal trend decreasing from
about 150 to 50 m (Fig. 7a). For $z < -100$ m below the seasonal stratification, vertical gradients
of macro-nutrients are large (Fig. 6b-d). Macro-nutrient values become approximately
independent of latitude at depths below $z < -500$ m. Dissolved iron profiles differ from macro-
nutrient profiles, notably in the upper layer near the surface (Fig. 6a). At some southern stations,
dissolved iron and to a lesser extent also phosphate, have relatively high concentrations closest
to the surface. These near-surface concentration increases suggest atmospheric sources, most
likely Saharan dust deposition (e.g., Rijkenberg et al., 2012).

As a function of latitude in the near-surface 'mixed' layer (Fig. 8), the vertical turbulent

fluxes of phosphate (representing the macro-nutrients, for graphical reasons, see the similarity
in profiles in Fig.6b-d) are found constant or insignificantly ($p > 0.05$) increasing (Fig. 8d).
Here, the mean eddy diffusivity values for the near-surface layer as presented in Fig. 5 are used
for computing the fluxes. It is noted that in this layer turbulent overturning (Figs 3b, 4b) is
larger and nutrients are mainly depleted (Fig. 6), except when replenished from atmospheric
sources in which case gradients reverse sign as in most DFe-profiles. Hereby, lateral diffusion
is not considered important. Nonetheless, macro-nutrients are seen to increase significantly
towards higher latitudes (Fig. 8b). We note that the vertical gradients in Fig. 8c, in which only
downgradient values are plotted, are very weak in general within the standard deviation of
measurements. The results in Fig. 8d are thus merely indicative, but they are consistent with
the results from deeper down presented below.

More importantly, the significant vertical turbulent fluxes of nutrients across the seasonal

pycnocline (Fig. 9) are found ambiguously or statistically independently varying with latitude
(Fig. 9d). Likewise, the vertical turbulent fluxes of dissolved iron and phosphate are marginally
constant with latitude across the more permanent stratification deeper down (Fig. 10). Nitrate
fluxes show the same latitudinal trend, with values around $10^{-6}$ mmol m$^{-2}$ s$^{-1}$. Overall, the
vertical turbulent nutrient fluxes across the seasonal and more permanent stratification resemble
those of the physical vertical turbulent mass flux, which is equivalent to the distribution of
turbulence dissipation rate and which is latitude-invariant (Fig. 5a).

**4 Discussion**

Practically, the upright positioning CTD while using an adaptation consisting of a custom-

made equal-surface inlet worked well to minimize ship-motion effects on variable flow-
imposed temperature variations. This improved calculated turbulence values from CTD-
observations in general and in near-homogeneous layers in particular. The indirect comparison
with turbulence values determined from previous microstructure profiler observations along the
same transect (Jurado et al., 2012) confirms the same trends, although occasionally turbulence
values were lower (to one order of magnitude in the present study). This difference in values
may be due to the time-lapse of 8 years between the observations, but more likely it is due to
inaccuracies in one or both methods. It is noted that any ocean turbulence observations cannot
be made better than to within a factor of two (Oakey, pers. comm.). In that respect, the standard
CTD with the here presented adaptation is a cheaper solution than additional microstructure
profiler observations. Although the general understanding, mainly amongst modellers, is that
the Thorpe length method overestimates diffusivity (e.g., Scotti, 2015; Mater and
Venayagamoorthy, 2015), this view is not shared amongst ocean observers (e.g., Gregg et al.,

2018). In the large parameter space of the high Reynolds number environment of the ocean, turbulence properties vary constantly, with an interminglement of convection and shear-induced turbulence at various levels. Given sufficient averaging, and adequate mean value parametrization, the Thorpe length method is not observed to overestimate diffusivity. This property of adequate and sufficient averaging yields similar mean parameter values in recent modelling results estimating a mixing coefficient near the classical bound of 0.2 in stationary flows for a wide range of conditions (Portwood et al., 2019). It is noted that diffusivity always requires knowledge of stratification to obtain a turbulent flux, and it is better to consider turbulence dissipation rate for intercomparison purposes. Nevertheless, future research may perform a more extensive comparison between Thorpe scale analysis data and deeper microstructure profiler data.

While our turbulence values are roughly similar to those of others transecting the NE-Atlantic over the entire water depth (Walter et al., 2005; Kunze et al., 2006), the focus in the present paper is on the upper 500 m because of its importance for upper-ocean marine biology. Our study demonstrates a significant decrease of stratification with increasing latitude and decreasing temperature that, however, does not lead to significant variation in turbulence values and vertical turbulent fluxes. Our direct estimates of the turbulent flux of nitrate into the euphotic zone are one to two orders of magnitude less than the previously estimated rate of nitrate uptake for the summer period. Our turbulent flux of nitrate values are of the same order of magnitude as reported by others (Cyr et al., 2015 and references therein). In particular, the Martin et al. (2010) study in the Northeast Atlantic Ocean (at 49°N, 16°W) reported similar vertical nutrient fluxes during summer, which provides confidence in the methods used. The same authors reported that the vertical nitrate flux into the euphotic zone was much lower than the rate of nitrate update at the time. To determine these nitrate uptake rates, they spiked water samples with a minimum of 0.5 μM nitrate, representing ~10% of the ambient nitrate concentration. In our study area, the ambient nitrate concentrations in the euphotic zone were much lower (see also Mojica et al., 2015), implying a higher relative importance of nitrate input to the overall uptake demand. Still, primary productivity in the oligotrophic euphotic zone, as

well as in the high latitude Atlantic, is mainly fueled by recycling (e.g., Gaul et al., 1999;
Achterberg et al., 2020) and the supply of new nutrients by turbulent fluxes, however small,
provides a welcome addition. Besides nutrient input resulting from vertical turbulent fluxes,
there is a role for latitudinal differences through the supply of nutrients by deep mixing events,
and depending on the location, also potential upwelling and lateral transport events.

We suggest that internal waves may drive the feed-back mechanism, participating in the

subtle balance between destabilizing shear and stable (re)stratification. Molecular diffusivity of
heat is about $10^{-7}$ $m^2$ $s^{-1}$ in seawater, and nearly always smaller than turbulent diffusivity in the
ocean. The average values of $K_z$ during our study were typically 100 to 1000 times larger than
molecular diffusivity, which implies turbulent diapycnal mixing drives vertical fluxes despite
the relatively slow turbulence compared to surface wave breaking. Depending on the gradient
of a substance like nutrients or matter, the relatively slow turbulence may not necessarily
provide weak fluxes $-K_z$d(substance)/dz into the photic zone. In the central North Sea, a
relatively low mean value of $K_z = 2\times10^{-5}$ $m^2$ $s^{-1}$ comparable to values over the seasonal
pycnocline here, was found sufficient to supply nutrients across the strong summer pycnocline
to sustain the entire late-summer phytoplankton bloom in near-surface waters and to warm up
the near-bottom waters by some 3°C over the period of seasonal stratification (van Haren et al.,
1999). There, the turbulent exchange was driven by a combination of tidal currents modified
by the stratification, shear by inertial motions driven by the Coriolis force (inertial shear) and
internal wave breaking. Such drivers are also known to occur in the open ocean, although to an
unknown extent.

The here observed (lack of) latitudinal trends of ε, $K_z$ and N yield approximately the same

information as the vertical trends in these parameters at all stations. In the vertical for z < -200
m, turbulence values of ε and $K_z$ weakly vary with stratification. This is perhaps unexpected
and contrary to the common belief of stratification hampering vertical turbulent exchange of
matter including nutrients. It is less surprising when considering that increasing stratification is
able to support larger shear. Known sources of destabilizing shear include near-inertial internal
waves of which the vertical length-scale is relatively small compared to other internal waves,
including internal tides (LeBlond and Mysak, 1978).
The dominance of inertial shear over shear by internal tidal motions (internal tide shear),
together with larger energy in the internal tidal waves, has been observed in the open-ocean,
e.g. in the Irminger Sea around 60°N (van Haren, 2007). The frequent atmospheric disturbances
in that area generate inertial motions and dominant inertial shear. Internal tides have larger
amplitudes but due to much larger length scales they generate weaker shear, than inertial
motions. Small-scale internal waves near the buoyancy frequency are abundant and may break
sparsely in the ocean interior outside regions of topographic influence. However, larger
destabilizing shear requires larger stable stratification to attain a subtle balance of 'constant'
marginal stability (van Haren et al., 1999). Not only storms but all geostrophic adjustments,
such as frontal collapse, may generate inertial wave shear also at low latitudes (Alford and
Gregg, 2001), so that overall latitudinal dependence may be negligible. If shear-induced
turbulence in the upper ocean is dominant it may thus be latitudinally independent (shallow
observations by Jurado et al., 2012; deeper observations in present study). There are no
indications that the overall open ocean internal wave field and (sub)mesoscale activities are
energetically much different across the mid-latitudes. If internal tide sources would have
dominated our observations, clear differences in turbulence dissipation rates would have been
found at our station near 48 °N (near the Porcupine Bank), for example, compared with those
at other stations.
Summarizing, our study infers that vertical nutrient fluxes did not vary significantly with
latitude and stratification. This suggests that predicted changes in the physical environment due
to global ocean warming have little effect on vertical turbulent exchange. Supposing that
enhanced warming leads to more stable stratification, more internal waves can be supported
(LeBlond and Mysak, 1978), which upon breaking can maintain the extent of vertical turbulent
exchange and thereby, for example, vertical nutrient fluxes. We thus hypothesize that, from a
physical environment perspective, in stratified oligotrophic waters the nutrient input from
deeper waters and corresponding summer phytoplankton productivity and growth are not
expected to change (much) with future global warming. We invite future observations and
numerical modelling to further investigate this suggestion and associated feed-back
mechanisms such as internal wave breaking.

*Data availability*. Data are available under doi/10.25850/nioz/7b.b.lb.

*Author contributions*. HvH analysed the data and drafted the paper. CPDB coordinated the
cruise. RM, MHvM and CPDB provided the nutrient and iron data. LJAG initiated the link of
disciplines in this study and stored the data sets. RG handled and operated the CTD-systems.
All authors contributed to the scientific discussion and edited the manuscript. All authors have
read and agreed to the published version of the manuscript.

*Competing interests*. The authors declare that they have no conflict of interest.

*Acknowledgements*. We thank the master and crew of the R/V Pelagia for their pleasant
contributions to the sea-operations. J. van Heerwaarden and R. Bakker made the CTD-
modification. We much appreciated the critical comments of the reviewers.

 APPENDIX A1

**517** **Modification of CTD pump-tubing to minimize RAM-effects**

**518**    The unique pump system on SeaBird Electronics (SBE) CTDs, foremost on their high-

**519**    precision full ocean depth shipborne and cable-lowered SBE911, minimizes the effects of flow

**520**    variations (and inversions) past its T-C sensors (SeaBird, 2012). This reduction in flow

**521**    variation is important because the T-sensor has a slower response than the C-sensor. As data

**522**    from the latter are highly temperature dependent, besides being pressure dependent, the precise

**523**    matching of all three sensors is crucial for establishing proper salinity and density

**524**    measurements, especially across rapid changes in any of the parameters. As flow past the T-

**525**    sensor causes higher measurement values due to friction at the sensor tip, flow-fluctuations are

**526**    to be avoided as they create artificial T-variations of about 1 mK s m$^{-1}$ (Larson and Pedersen,

**527**    1996).

**528**    However, while the pump itself is one thing, its tubing needs careful mounting as well, with

**529**    in- and outlet at the same depth level (Sea-Bird, 2012). This is to prevent ram pressure $P = \rho U^2$,

**530**    for density $\rho$ and flow speed U. Unfortunately, the SBE-manual shows tubing of different

**531**    diameters, for in- and outlet. Different diameter tubing leads to velocity fluctuations of $\pm 0.5$ m

**532**    s$^{-1}$ past the T-sensor, as was concluded from a simple experiment by van Haren and Laan

**533**    (2016). The flow speed variations induce temperature variations of $\pm 0.5$ mK and are mainly

**534**    detectable in weakly stratified waters such as in the deep ocean, but also near the surface as

**535**    observed in the present data. Using tubes of the same diameter opening remedied most of the

**536**    effect, but only if the surface of the tube-opening is perpendicular to the main CT-motion as in

**537**    a vertically mounted CTD. If it is parallel to the main motion as in a horizontally mounted CTD,

**538**    the effect was found to be adverse. The make-shift onboard experiment in van Haren and Laan

**539**    (2016) has now been cast into a better design (Fig. A1), of which the first results are presented

**540**    in this paper.

APPENDIX A2

**PDFs of vertically averaged dissipation rate in comparison with latitudinal trends**

Ocean turbulence dissipation rate generally tends to a nearly log-normal distribution (e.g., Pearson and Fox-Kemper, 2018), so that the probability density function (PDF) of the logarithm of $\varepsilon$-values is normally distributed and can be described by the first two moments, the mean and its standard deviation. It is seen in Fig. A2a that the overall distribution of all present data indeed approaches lognormality, despite the relatively large length-scale used in the computations (cf., Yamazaki and Lueck, 1990). When the data are split into the three depth levels as in Fig. 5a, it is seen that $\varepsilon$ in the upper z > -15 m layer is not log-normally distributed due to a few outlying high values confirming an ocean state dominated by a few turbulence bursts (Moum and Rippeth, 2009), whereas $\varepsilon$ in the deeper more stratified layers is nearly log-normally distributed.

When we compare the mean and standard deviations of the distributions with the extreme values of the latitudinal trends as computed for Fig. 5a it is seen that for none of the three depth levels the extreme values are found outside one standard deviation from the mean value. In fact, for deeper stratified waters the extreme values of the trends are found very close to the mean value. It is concluded that the mean dissipation rate does not show a significant trend with latitude, at all depth levels. The same exercise yields extreme buoyancy frequency values lying outside one standard deviation from the mean values for well-stratified waters, from which we conclude that stratification significantly decreases with latitude. This is inferable from Fig. 5c by investigating the spread of mean values around the trend line.

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

**Figure 1**. Bathymetry map of the Northeast Atlantic Ocean based on the 9.1 ETOPO-1 version
of satellite altimetry-derived data by Smith and Sandwell (1997). The numbered circles
indicate the CTD stations, at station 17 (x) no turbulence parameter, only nutrient sampling
was done. At stations 1 and 2 no DFe-samples were taken, at station 18 no nutrient-samples
were taken. Depth contours are at 2500 and 5000 m.

**Figure 2**. Test of effective removal of ship motions in CTD-data after pump in- and outlet
modification. Nearly raw 24 Hz sampled downcast data obtained from northern station 32
(cast 9). Short example time series for the 20-m depth range [10, 30] m. (a) Detrended
pressure (blue) and its (negative signed) first time derivative -dp/dt, 2-dbar-smoothed
(purple). (b) Detrended temperature. (c) Moderately smoothed (~30 degrees of freedom;
dof) spectra of data from the 5 to 500 m depth range. (d) Moderately smoothed (40 dof)
coherence between dp/dt and T from c., with dashed line indicating the 95% significance
level. (e) Corresponding phase difference.

**Figure 3**. Upper 500 m of turbulence characteristics computed from downcast density anomaly
data applying a threshold of $7 \times 10^{-5}$ kg m$^{-3}$. Northern station 29, cast 2. (a) Unordered, 'raw'
profile of density anomaly referenced to the surface. (b) Overturn displacements following
reordering of the profiles in a. Slopes ½ (solid lines) and 1 (dashed lines) are indicated. (c)
Logarithm of dissipation rate computed from the profiles in a., r.m.s. calculated over 7 m
intervals. We use the mathematics expression 'lg' for the 10-base logarithm, as given in
the ISO 80000 specification. (d) As c., but for eddy diffusivity. (e) Logarithm of buoyancy
frequency computed after reordering the profiles of a.

**Figure 4**. As Fig. 3, but for a southern station. Upper 500 m of turbulence characteristics
computed from downcast density anomaly data applying a threshold of $7 \times 10^{-5}$ kg m$^{-3}$.
Southern station 3, cast 4. (a) Unordered, 'raw' profile of density anomaly referenced to
the surface. (b) Overturn displacements following reordering of the profiles in a. Slopes ½
(solid lines) and 1 (dashed lines) are indicated. (c) Logarithm of dissipation rate computed
from the profiles in a., r.m.s. calculated over 7 m intervals. (d) As c., but for eddy
diffusivity. (e) Logarithm of buoyancy frequency computed after reordering the profiles of
a.

**Figure 5**. Summer 2017 latitudinal transect along $17\pm5°W$ of turbulence values for upper 15 m
averages (green) and averages between $-100 < z < -25$ m (blue, seasonal pycnocline) and -
$500 < z < -100$ m (black, more permanent pycnocline) from short yoyos of 3 to 6 CTD-
casts. Values are given per cast (o) and station average (heavy circle with x; the size
corresponds with ±the standard error for turbulence parameters). (a) Logarithm of
dissipation rate. (b) Logarithm of diffusivity. (c) Logarithm of buoyancy frequency (the
small symbols have the size of ±the standard error). (d) Hour of sampling after sunrise.

**Figure 6**. Upper 500 m profiles for stations at three latitudes. (a) Density anomaly referenced
to the surface, including profiles from Fig. 3a and 4a. (b) Nitrate plus nitrite. (c) Phosphate.
(d) Silicate. (e) Dissolved iron.

**Figure 7**. Latitudinal transect of near-surface layers and wind conditions measured at stations
during the observational survey. (a) Mixed layer depth (x) and euphotic zone (o). (b) Wind
speed. (c) Wind direction.

**Figure 8**. Latitudinal transect of near-surface nutrient concentrations. (a) Dissolved iron
measured at depths indicated. Missing values reflect not all depths were sampled. (b)
Nitrate plus nitrite (red) and phosphate (blue, scale times 10) measured at depths indicated
in a. (c) Logarithm of (very weak within standard deviations of measurements) vertical
gradients of dissolved iron in a. (o-red) and phosphate in b. (x-blue). Only downgradient
values are shown, which excludes several $PO_4$- and nearly all DFe-gradient values due to
near-surface increased values (*cf*. Fig. 6e, 32°N profile). (d). Upward vertical turbulent

fluxes of phosphate concentration gradients in c. using average surface $K_z$ from Fig. 5b,

valid for depth average (here, ~17 m) of depths in a.


**Figure 9**. As Fig. 8, but for -100 < z < -25 m, with fluxes for ~62 m in d.


**Figure 10**. As Fig. 8, but for -600 (few nutrients sampled at 500) < z < -100 m, with fluxes for

~350 m in d.


**Fig. A1**. SBE911 CTD-pump in- and outlet modification following the findings in van Haren

and Laan (2016). (a) The T- and C-sensors clamped together with a structure holding in-

and outlet pump-tubing of exactly the same diameter, separated at 0.3 m distance in the

horizontal plane. (b) The modification of a. mounted in the CTD-frame.

**Fig. A2**. Probability Density Functions of logarithm of vertically averaged dissipation rate in

comparison with latitudinal trend extreme values. (a) Distribution as a function of latitude

for all data. (b) As a, but for the upper 15 m averages only. The mean value is given by the

vertical purple line, with the horizontal line indicating +/- 1 standard deviation. The vertical

light-blue lines indicate the best-fit value of the trend for 30° and 63°N. (c) As b, but for

averages between -100 < z < -25 m. (d) As c, but for averages between -500 < z < -100 m.

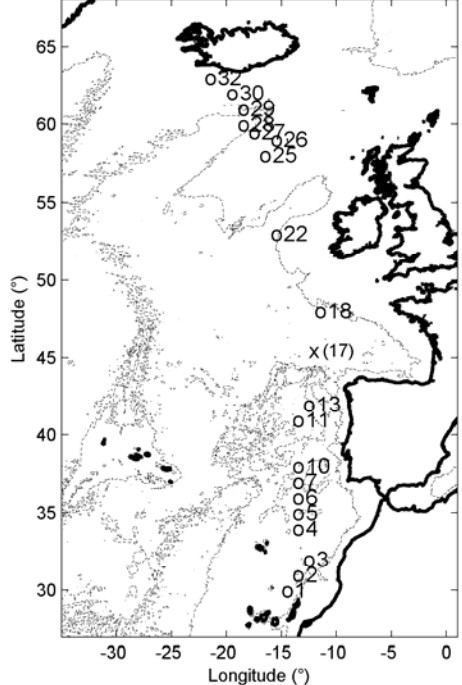

**Figure 1**. Bathymetry map of the Northeast Atlantic Ocean based on the 9.1 ETOPO-1
version of satellite altimetry-derived data by Smith and Sandwell (1997). The numbered
circles indicate the CTD stations, at station 17 (x) no turbulence parameter, only nutrient
sampling was done. At stations 1 and 2 no DFe-samples were taken, at station 18 no
nutrient-samples were taken. Depth contours are at 2500 and 5000 m.

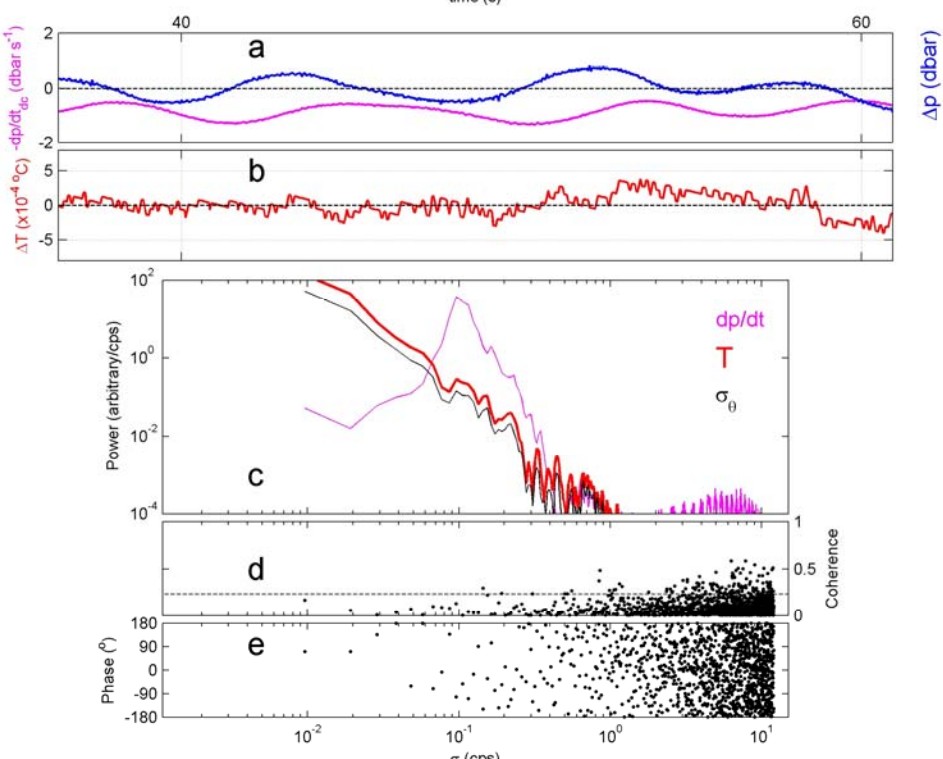

**Figure 2**. Test of effective removal of ship motions in CTD-data after pump in- and outlet
modification. Nearly raw 24 Hz sampled downcast data obtained from northern station 32
(cast 9). Short example time series for the 20-m depth range [10, 30] m. (a) Detrended
pressure (blue) and its (negative signed) first time derivative -dp/dt, 2-dbar-smoothed
(purple). (b) Detrended temperature. (c) Moderately smoothed (~30 degrees of freedom;
dof) spectra of data from the 5 to 500 m depth range. (d) Moderately smoothed (40 dof)
coherence between dp/dt and T from c., with dashed line indicating the 95% significance
level. (e) Corresponding phase difference.

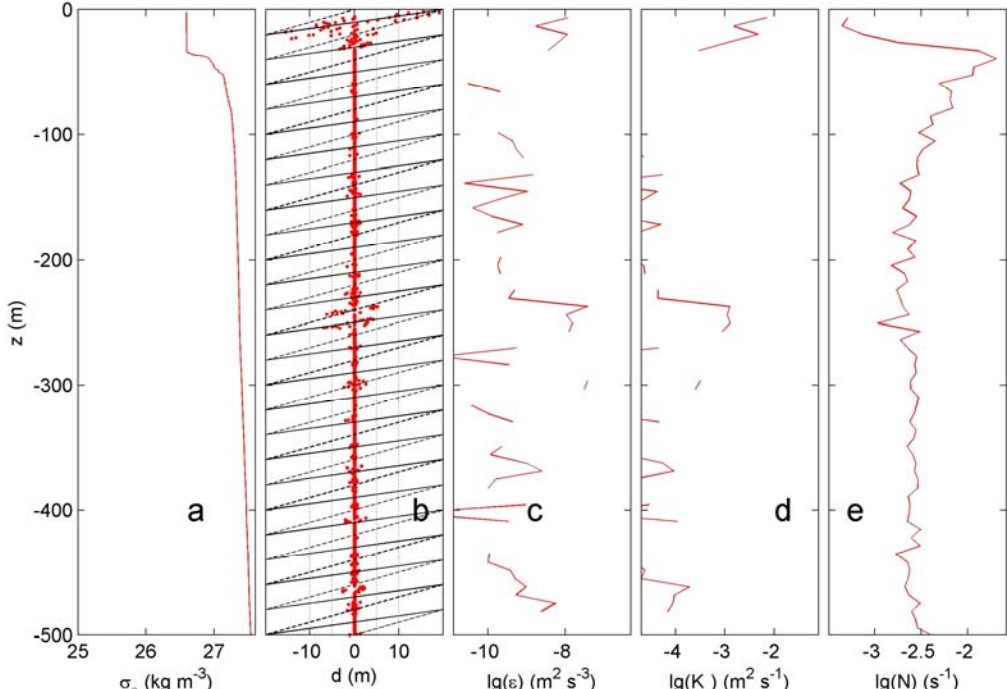

**Figure 3**. Upper 500 m of turbulence characteristics computed from downcast density
anomaly data applying a threshold of $7 \times 10^{-5}$ kg m$^{-3}$. Northern station 29, cast 2. (a)
Unordered, 'raw' profile of density anomaly referenced to the surface. (b) Overturn
displacements following reordering of the profiles in a. Slopes ½ (solid lines) and 1
(dashed lines) are indicated. (c) Logarithm of dissipation rate computed from the profiles
in a., r.m.s. calculated over 7 m intervals. We use the mathematics expression 'lg' for the
10-base logarithm, as given in the ISO 80000 specification. (d) As c., but for eddy
diffusivity. (e) Logarithm of buoyancy frequency computed after reordering the profiles
of a.

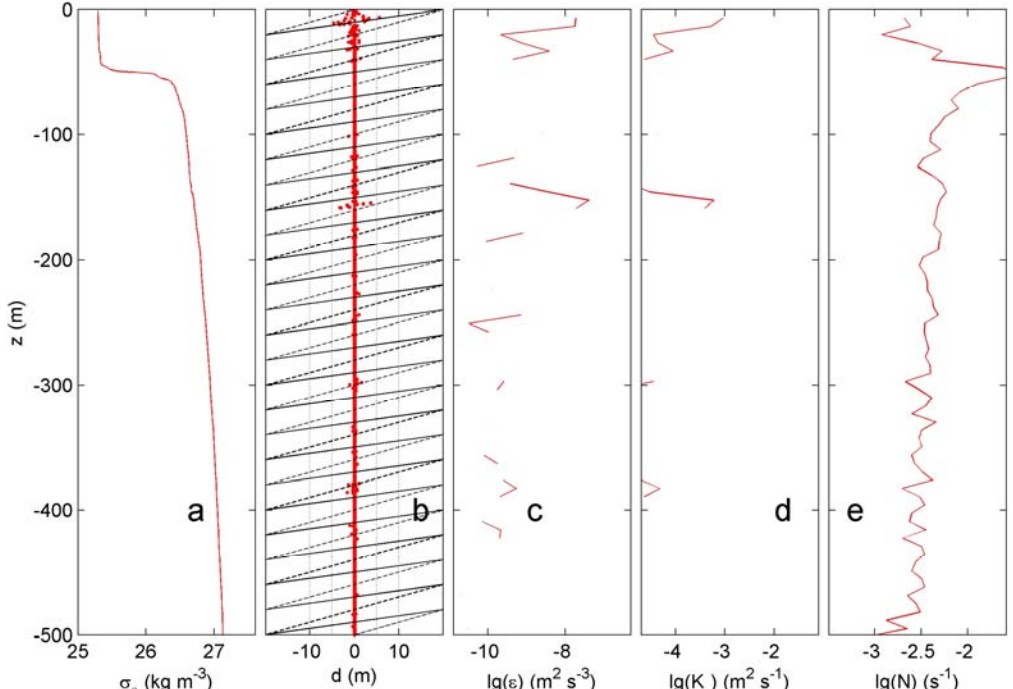

**Figure 4**. As Fig. 3, but for a southern station. Upper 500 m of turbulence characteristics
computed from downcast density anomaly data applying a threshold of $7 \times 10^{-5}$ kg m$^{-3}$.
Southern station 3, cast 4. (a) Unordered, 'raw' profile of density anomaly referenced to
the surface. (b) Overturn displacements following reordering of the profiles in a. Slopes ½
(solid lines) and 1 (dashed lines) are indicated. (c) Logarithm of dissipation rate computed
from the profiles in a., r.m.s. calculated over 7 m intervals. (d) As c., but for eddy
diffusivity. (e) Logarithm of buoyancy frequency computed after reordering the profiles of
a.

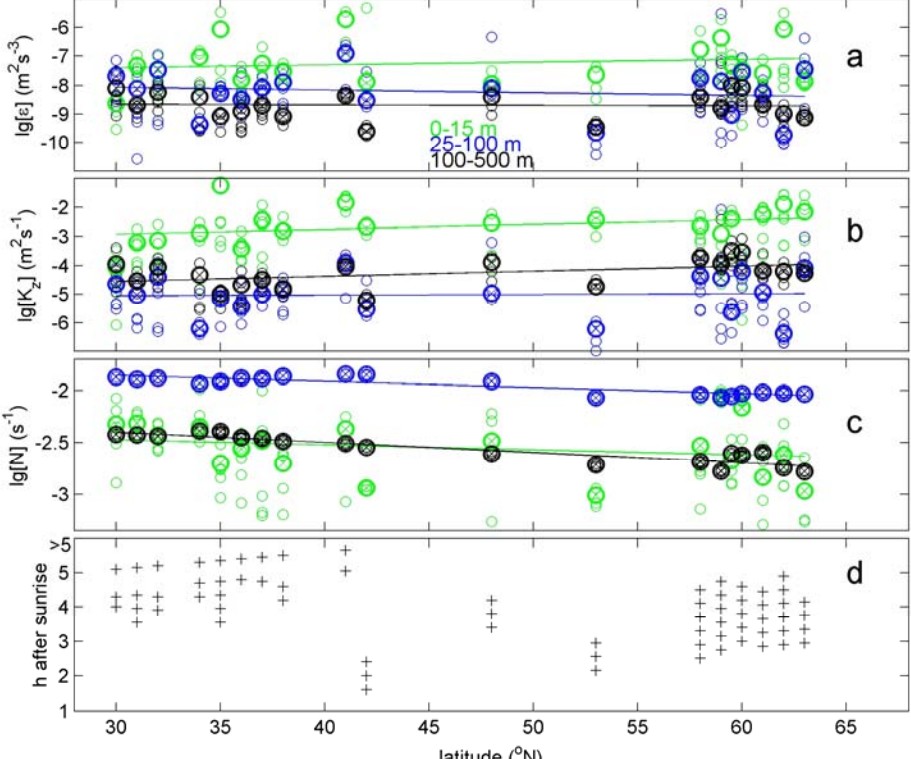

**Figure 5**. Summer 2017 latitudinal transect along 17±5°W of turbulence values for upper 15 m averages (green) and averages between -100 < z < -25 m (blue, seasonal pycnocline) and -500 < z < -100 m (black, more permanent pycnocline) from short yoyos of 3 to 6 CTD-casts. Values are given per cast (o) and station average (heavy circle with x; the size corresponds with ±the standard error for turbulence parameters). (a) Logarithm of dissipation rate. (b) Logarithm of diffusivity. (c) Logarithm of buoyancy frequency (the small symbols have the size of ±the standard error). (d) Hour of sampling after sunrise.

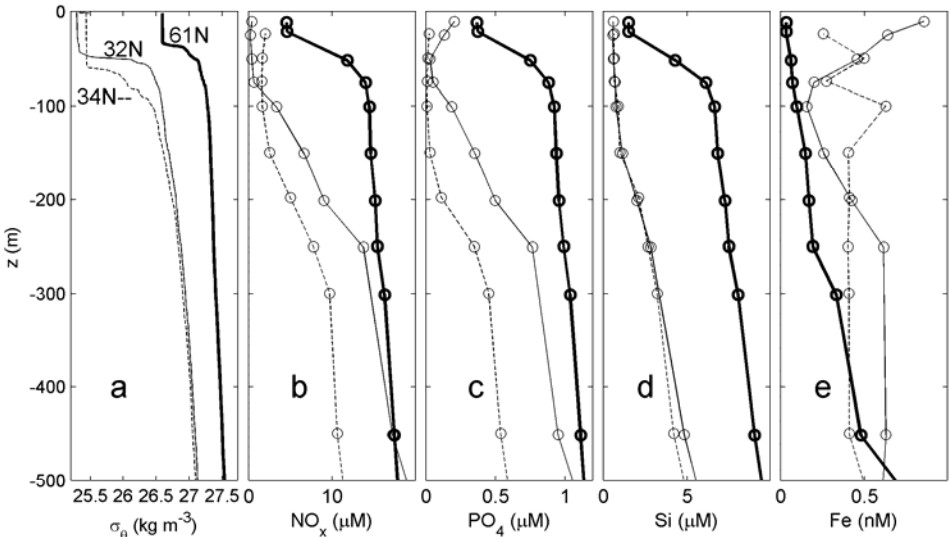

**Figure 6**. Upper 500 m profiles for stations at three latitudes. (a) Density anomaly
referenced to the surface, including profiles from Fig. 3a and 4a. (b) Nitrate plus nitrite. (c)
Phosphate. (d) Silicate. (e) Dissolved iron.


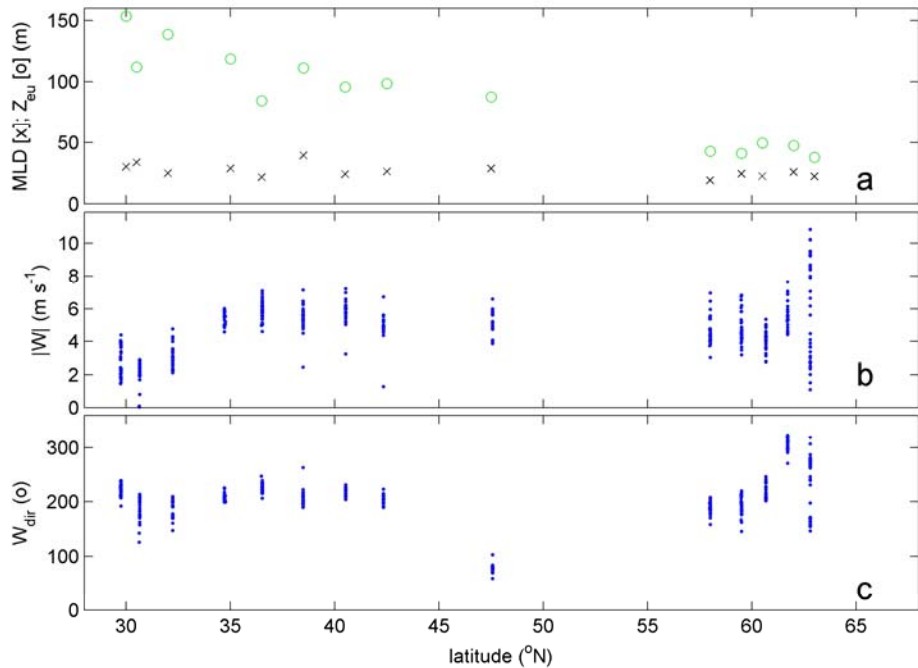

**Figure 7**. Latitudinal transect of near-surface layers and wind conditions measured at
stations during the observational survey. (a) Mixed layer depth (x) and euphotic zone (o).
(b) Wind speed. (c) Wind direction.

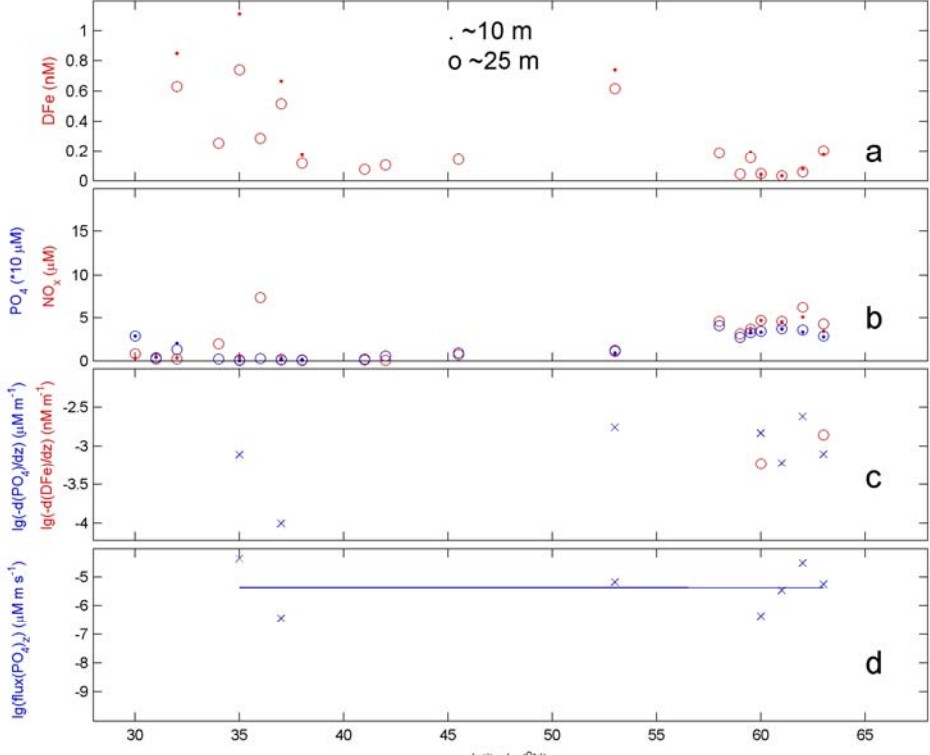

**Figure 8**. Latitudinal transect of near-surface nutrient concentrations. (a) Dissolved iron
measured at depths indicated. Missing values reflect not all depths were sampled. (b)
Nitrate plus nitrite (red) and phosphate (blue, scale times 10) measured at the depths
indicated in a. (c) Logarithm of (very weak within standard deviations of measurements)
vertical gradients of dissolved iron in a. (o-red) and phosphate in b. (x-blue). Only
downgradient values are shown, which excludes several $PO_4$- and nearly all DFe-gradient
values due to near-surface increased values (*cf*. Fig. 6e, 32°N profile). (d). Upward vertical
turbulent fluxes of phosphate concentration gradients in c. using average surface $K_z$ from
Fig. 5b, valid for the depth average (here, ~17 m) of depths in a.

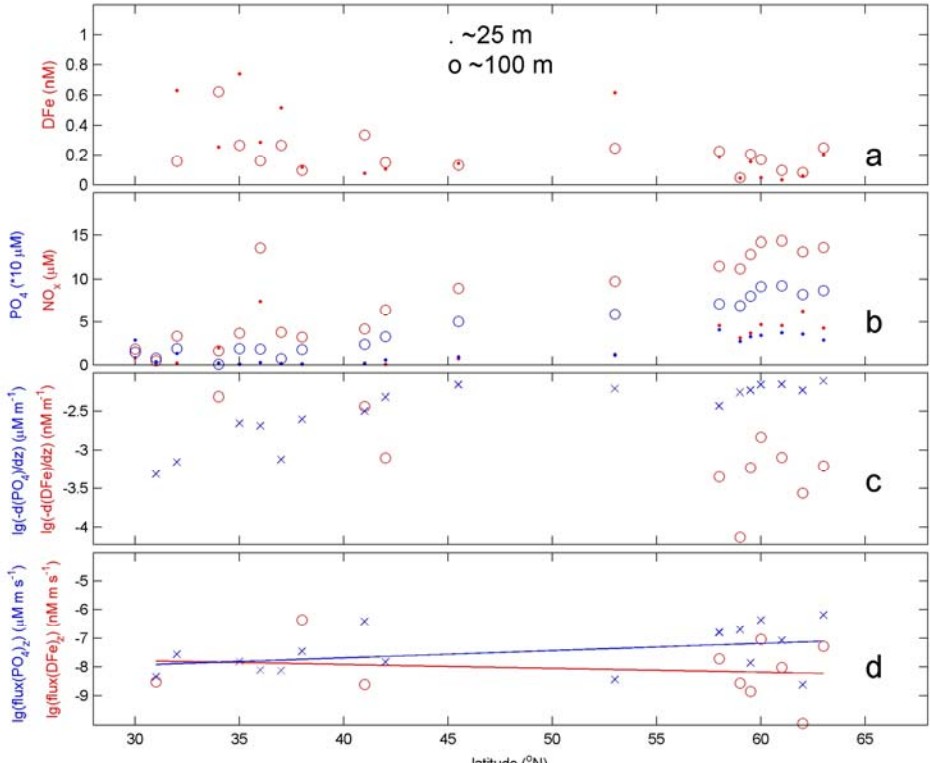

**Figure 9**. As Fig. 8, but for -100 < z < -25 m, with fluxes for ~62 m in d.


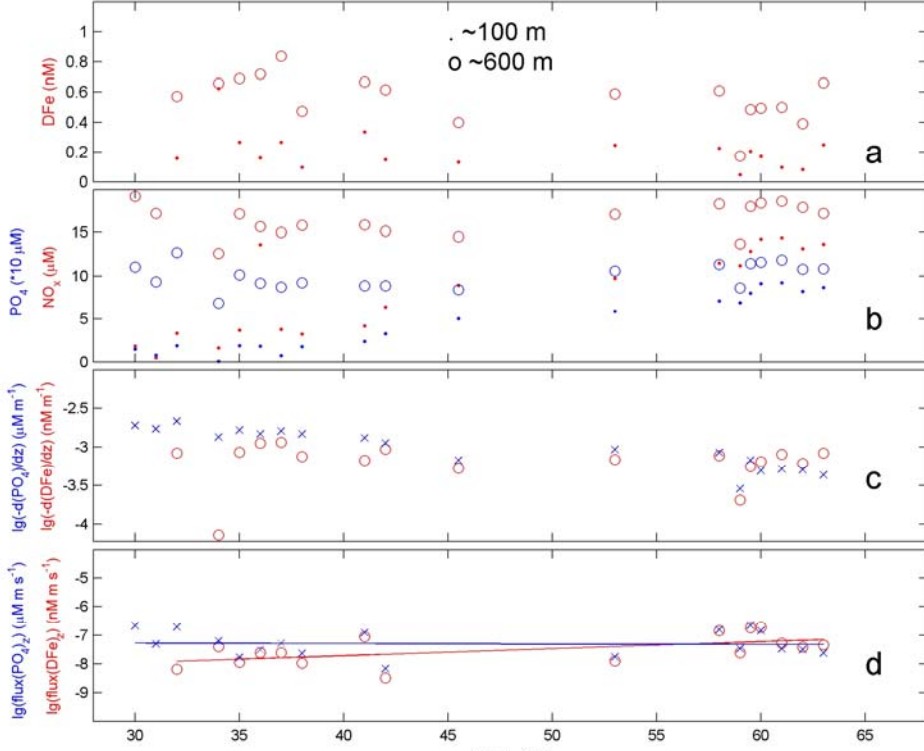

**Figure 10**. As Fig. 8, but for -600 (few nutrients sampled at 500) < z < -100 m, with fluxes
for ~350 m in d.

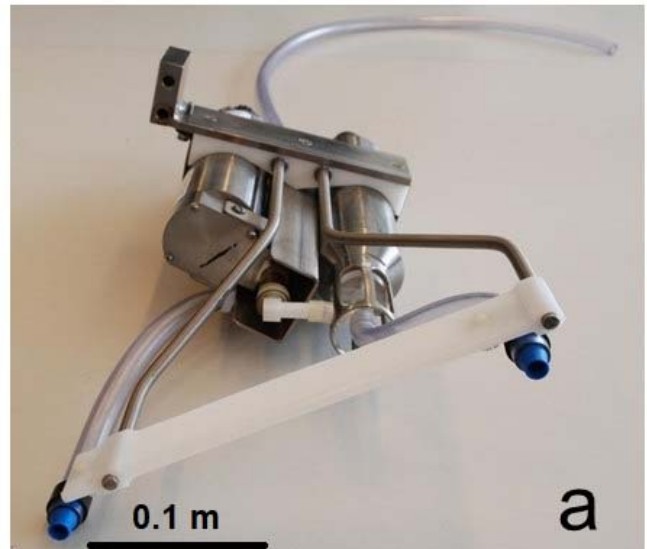

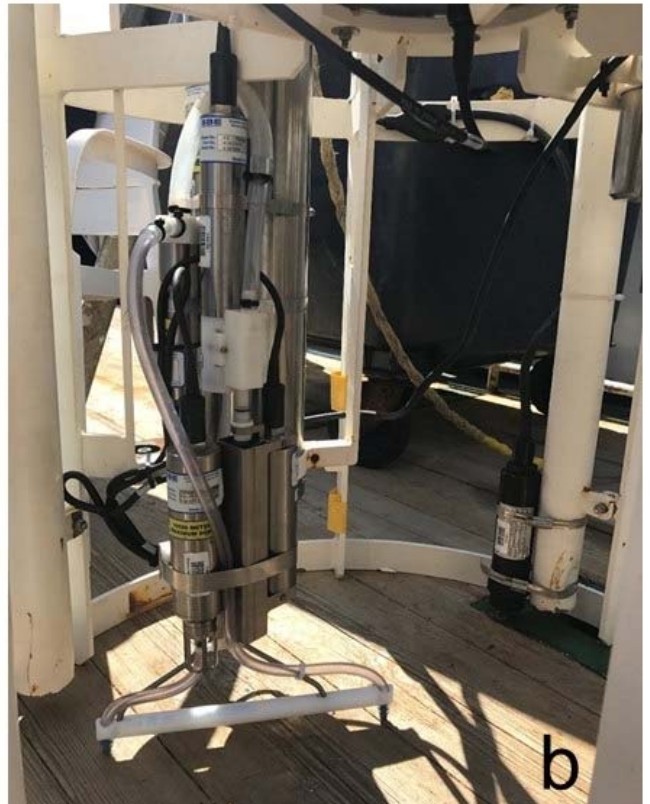

**Fig. A1**. SBE911 CTD-pump in- and outlet modification following the findings in van
Haren and Laan (2016). (a) The T- and C-sensors clamped together with a structure holding
in- and outlet pump-tubing of exactly the same diameter, separated at 0.3 m distance in the
horizontal plane. (b) The modification of a. mounted in the CTD-frame.

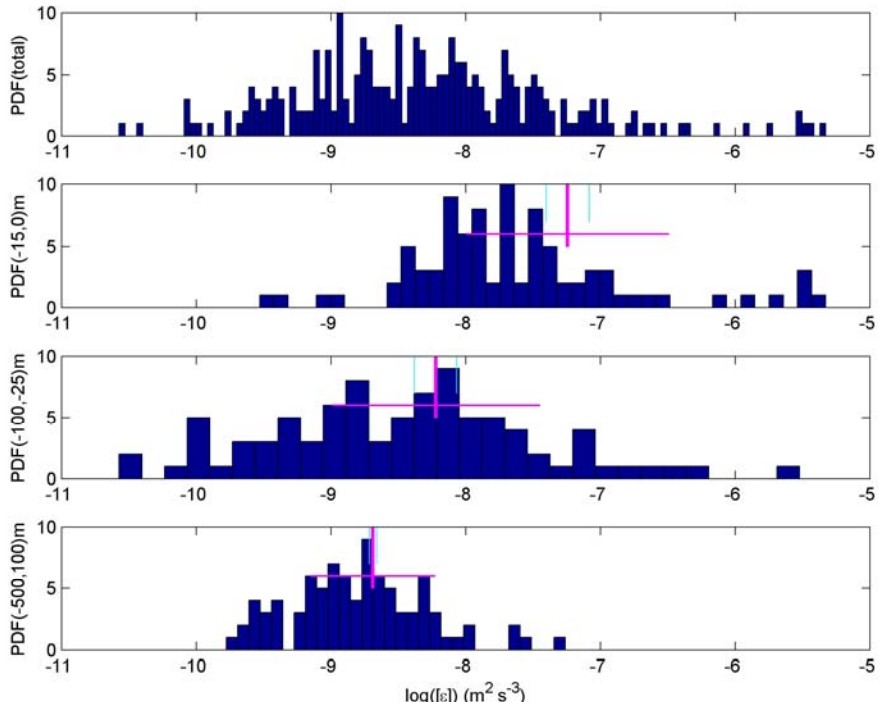

**Fig. A2**. Probability Density Functions of logarithm of vertically averaged dissipation rate
in comparison with latitudinal trend extreme values. (a) Distribution as a function of
latitude for all data. (b) As a, but for the upper 15 m averages only. The mean value is given
by the vertical purple line, with the horizontal line indicating +/- 1 standard deviation. The
vertical light-blue lines indicate the best-fit value of the trend for 30° and 63°N. (c) As b,
but for averages between -100 < z < -25 m. (d) As c, but for averages between -500 < z < -
100 m.
