# Peer review of "Diapycnal mixing across the photic zone of the"

_Ocean Science, 2020_

## Referee Comment (RC1) · Anonymous Referee #1 · 4 Sep 2020

Review for manuscript # os-2020-73 "Diapycnal mixing across the photic zone of the NE-Atlantic" by van Haren et al.

Formal review:

The authors discuss dissipation rates of turbulent kinetic energy, eddy diffusivities and vertical turbulent nutrient fluxes inferred from upper-ocean hydrographic and nutrient data taken during a cruise on a transect from 60°N to 30°N along about 17°W in the North Atlantic. Inferred eddy diffusivities and vertical turbulent nutrient fluxes in the upper thermocline (<500m depth) did not vary with latitude. However, from south to north stratification in the upper thermocline weakened by a factor of 5. The authors claim that the lack of correspondence between turbulent mixing and stratification (temperature) suggest that nutrient availability for phytoplankton in the euphotic surface waters may

not be affected by global warming.

While this paper is fairly well written and addresses scientifically relevant question such as an advancing quantitative understanding of the role of mixing in sustaining biological production in the near surface layers of the ocean, the current version of the manuscript has major deficiencies. In particular, I find that the results presented in the manuscript are not sufficient to support the authors' interpretations and conclusions. Furthermore, a statistical analysis of uncertainties inherent to the results needs to be added.

Major concerns

Personally, I per se agree with the statement that climate warming and associated increase of upper-ocean stratification will not necessarily lead to a decrease of turbulent mixing in the thermocline or a decrease of vertical turbulent nutrient fluxes. Certainly, there are also arguments that support an enhanced energy flux into internal waves due to increasing stratification (which has also been suggest by several previous publications, e.g. DeCarlo et al., 2015). However, to me, the data analysis presented here does not permit to draw any conclusions on this issue. This is because (1) the data is inadequately resolving average mixing quantities. Turbulent mixing in the ocean exhibits a near log-normal frequency distribution and elevated mixing events occur infrequently. However, these elevated mixing events are dominantly responsible for the vertical turbulent fluxes of solutes in the ocean. The 60+ profiles (I am guessing here as no numbers are provided in the manuscript) that may represent turbulence conditions over a period of 3 to 4 hours at the 15 to 20 individual stations are certainly inadequate to draw any conclusions on average turbulence quantities at different latitudes. The variability of turbulent mixing is also reflected by (2) the individual estimates of vertical turbulent nutrient fluxes available from the limited individual stations along the transect. Fluxes vary by three orders of magnitude (Figures 7, 8, 9). Again, an analysis of their statistical uncertainty would show the ambiguity of any trend analysis. Finally, (3) I cannot approve the approach chosen here as a whole. Comparing the strength of upper thermocline mixing at different latitudes cannot lead to any conclusions on
local changes of the strength of turbulent mixing e.g. due to locally increasing stratification. The regions where measurements were taken combine very different external forcing and internal wave environments making it impossible to relate mixing strength to a single parameter.

As a revision strategy, I would advise the authors to remove the discussion on mixing and nutrient fluxes in a changing climate from the manuscript. Instead, the focus could be shifted to a detailed discussion of an upper-ocean nutrient budget including statistical uncertainties and a comparison to the net community production.

Some specific comments

Line 294 – 299, discussion of nutrient fluxes in the mixed layer and Fig. 7. I find the discussion of macro-nutrient fluxes in the mixed layer erroneous. First of all, vertical gradients of macro-nutrients are mostly insignificant. Macro-nutrient concentrations determined by a QuAAtro autoanalyser usually have accuracies of 0.1mM if CRM standards were used (please add details of uncertainties inherent to the nutrient concentrations to the methods section). To me, the differences between macro-nutrient concentrations measured at 10m and 25m depth are mostly smaller than measurement uncertainties.

Line 300 – 303, discussion of nutrient fluxes below the mixed layer. As stated in the above, individual estimates of nutrient fluxes vary by three orders of magnitude and a statement about how the nutrient fluxes vary with latitude (i.e. with stratification) is inadequate. What may be interesting to the reader is the magnitude of average regional fluxes that could be compared to previous estimates (see e.g. Cyr et al., 2015). Presented results should also include nitrate/nitrite fluxes as the relative vertical turbulent fluxes of reactive nitrogen species and phosphorous could be of interest to a broader scientific community.

Literature

Cyr, F., D. Bourgault, P. S. Galbraith, and M. Gosselin (2015), Turbulent nitrate fluxes in
the Lower St. Lawrence Estuary, Canada, J. Geophys. Res. Oceans, 120, 2308–2330, doi:10.1002/2014JC010272.

DeCarlo, T. M., K. B. Karnauskas, K. A. Davis,and G.T.F. Wong (2015), Climate modulates internal wave activity in the Northern South China Sea, Geophys. Res.Lett., 42, 831–838, doi:10.1002/2014GL062522.

---

## Referee Comment (RC2) · Anonymous Referee #2 · 6 Sep 2020

The manuscript titled "Diapycnal mixing across the photic zone of the NE-Atlantic" by Haren et al. quantified the upper ocean nutrient flux using a custom modified CTD and nutrient measurements at discrete depths from a latitudinal transect along $17\pm5°$W between 30 and 62°N in summer. The authors observed no increase in vertical mixing or diapycnal nutrient flux from south to north, where the temperature increased. Further, they opined that nutrient supply by diapycnal flux to the euphotic zone might not be affected by the physical process of global warming. It is a well-written manuscript and presents an interesting take on the ocean biophysical coupling in the global warming scenario. However, I feel that the authors jumped into a conclusion without providing enough evidence to support their say. Hence I recommend major revision.

Major Comments

[Figure]

L63-96 The introduction needs a more general introduction to the oceanography of the region. Especially knowledge of bathymetry, background internal wave field, eddy kinetic energy, and wind conditions during summer.

L123 In the Thorpe length calculation section, please mention the lowering speed of the CTD. A slow lowering can resolve overturns efficiently. In the mixed layer, the Thorpe method will consider it as a large overturn.

How you will justify the validity of diffusivity within the mixed layer, where $N^2$ is weak. A brief discussion on lowering speed of CTD and justification for the diffusivity within the mixed layer will give clarity to the reader.

L256-258 Substantiate the surface cooling and internal wave breaking using data.

L264-265 I could not understand this sentence.

L284-286 The nutrient flux depends on the eddy diffusivity and the nutrient concentration gradient, which changes dramatically with depth. The nutrient fluxes thus may vary with two-or-three orders difference. In the manuscript, nutrient flux is calculated using a low-resolution profile of nutrients. Does this discrete measurement introduce bias to the flux calculation?

What is the typical depth of the euphotic zone in the study region?

L318-320 General understanding is that the Thorpe length method overestimates the diffusivity (Mater and Venayagamoorthy 2015; Alberto Scotti 2015).

L328-329 Here you can add a detailed discussion on how internal waves can be a feedback mechanism to counteract the suppression of mixing by increased stratification.

L344-364 Authors need to provide data evidence to prove that Internal wave energy/eddy kinetic energy is more in Northern stations, and thus, the relatively increased stratification (compared to south) could not suppress the diapycnal flux of nutrients to the euphotic zone from deeper depths.

This will give the readers a better understanding of the lack of correspondence between temperature /stratification and diapycnal flux with latitude.

One could employ the GM spectrum calculated using gridded Historical data sets (ARGO) to give an idea on the background Internal wave energy. However, I won't insist on doing this analysis.

A discussion on the meteorological conditions during the observation period is also warranted. What if the southern stations were characterized with anomalously calm weather that mixing was inactive and became comparable to the northern stations.

References

Scotti, A. (2015). Biases in Thorpe-scale estimates of turbulence dissipation. Part II: energetics arguments and turbulence simulations. Journal of Physical Oceanography, 45(10), 2522-2543.

Mater, B. D., Venayagamoorthy, S. K., St. Laurent, L., & Moum, J. N. (2015). Biases in Thorpe-scale estimates of turbulence dissipation. Part I: Assessments from large-scale overturns in oceanographic data. Journal of Physical Oceanography, 45(10), 2497-2521.

---

## Referee Comment (RC3) · Anonymous Referee #3 · 20 Sep 2020

This is an observational study of diapycnal mixing and the corresponding nutrient flux in the upper ocean across a quasi-latitudinal transect in the Northeast Atlantic Ocean. The data cover a rather long distance from 30deg.N to 62deg.N. The measurements were mainly temperature and conductivity profiles (from which the density or potential density profiles were obtained) with a carefully modified CTD system, and the turbulent kinetic energy dissipation rate and the diapycnal diffusivity were estimated based on the overturning (Thorpe) scale analysis. In general, the methodology of the analysis is reasonable, and useful information on turbulent mixing characteristics along the transect is obtained. However, I cannot recommend this manuscript for publication in the present form due to the major concerns as detailed in the following.

First of all, I find the major point that the authors try to make (i.e., "nutrient availability

for phytoplankton in the euphotic surface waters may not be affected by the physical process of global warming") is not convincing at all. For me, the point is not even relevant to what the data have shown. Obviously, the exact response of the upper ocean to global warming could be rather complicated, and I do agree with the authors that the global warming may not necessarily lead to a change in vertical turbulent exchange, but the results presented in the manuscript are by no means evidence for this. One may expect to see clear trend of upper ocean mixing (and corresponding material fluxes) at a certain location under continuing warming, but in such a large region covering more than 30 degrees, the underlying dynamics controlling diapycnal mixing could be very different from place to place, thus the spatial difference in mixing seen along the transect cannot be simply taken as a result of the difference in stratification (or "warming" by solar radiation).

More technically, although I do appreciate the authors' efforts in estimating turbulence and mixing characteristics from carefully conducted CTD measurements (via overturning scale analysis), I cannot be convinced by the subtle mixing (and flux) variability revealed by their estimates. As well acknowledged by the authors, even with the microstructure measurements one cannot expect to get an estimate with insignificant uncertainty. I agree that the overturning (Thorpe) scale analysis could be very useful in getting a rough estimate of mixing intensity when more direct measurements are not available, but using it to reveal subtle spatial (or temporal) variability could be misleading. For this purpose, direct microstructure measurements are certainly much more reliable. On the other hand, ocean turbulence is certainly a stochastic process with both significant dynamical variability (which could be taken as deterministic linked to certain dynamical processes generating turbulence) and intermittency. As such, for the purpose of evaluating spatial variability of turbulent mixing, one should look at turbulence statistics. How many data points are used to get the reported averages? How does the PDF in each corresponding depth range look like? What are the confidence intervals of the reported averages? Are the noted differences/variabilities really significant?

To conclude, I agree that the reported analysis gives useful information about mixing characteristics along the sampled transect, but without clear information of the underlying mechanisms and robust constraint on the reliability and significance of the reported mixing variability, one cannot be led to the points that the authors try to make. In particular, the results presented in the manuscript do not seem to lend any support to the authors' argument on the global warming impact on upper ocean mixing and nutrient flux trend. The authors may choose to simply emphasize their mixing estimates from the overturning scale analysis, with clear indication of the underlying uncertainties.

---

## Author Comment (AC1) · 27 Oct 2020

*>>>We are grateful for the comments on our manuscript from the reviewer. We feel that this new version of the paper is much stronger as the result of the comments we received on the original manuscript. We have addressed all of the comments and have detailed our response to specific comments below. Our response to each comment is bulleted and in italics below the relevant comment behind>>>*

**Anonymous Referee #1**

Review for manuscript # os-2020-73 "Diapycnal mixing across the photic zone of the NE-Atlantic" by van Haren et al.

Formal review:
The authors discuss dissipation rates of turbulent kinetic energy, eddy diffusivities and vertical turbulent nutrient fluxes inferred from upper-ocean hydrographic and nutrient data taken during a cruise on a transect from 60_N to 30_N along about 17_W in the North Atlantic. Inferred eddy diffusivities and vertical turbulent nutrient fluxes in the upper thermocline (<500m depth) did not vary with latitude. However, from south to north stratification in the upper thermocline weakened by a factor of 5. The authors claim that the lack of correspondence between turbulent mixing and stratification (temperature) suggest that nutrient availability for phytoplankton in the euphotic surface waters may not be affected by global warming.
While this paper is fairly well written and addresses scientifically relevant question such as an advancing quantitative understanding of the role of mixing in sustaining biological production in the near surface layers of the ocean, the current version of the manuscript has major deficiencies. In particular, I find that the results presented in the manuscript are not sufficient to support the authors' interpretations and conclusions. Furthermore, a statistical analysis of uncertainties inherent to the results needs to be added.
*>>>We thank the reviewer for the appreciation. We have now attempted to substantiate support for our interpretations. Uncertainties to the results are further explained.*

Major concerns
Personally, I per se agree with the statement that climate warming and associated increase of upper-ocean stratification will not necessarily lead to a decrease of turbulent mixing in the thermocline or a decrease of vertical turbulent nutrient fluxes. Certainly, there are also arguments that support an enhanced energy flux into internal waves due to increasing stratification (which has also been suggest by several previous publications, e.g. DeCarlo et al., 2015). However, to me, the data analysis presented here does not permit to draw any conclusions on this issue. This is because (1) the data is inadequately resolving average mixing quantities. Turbulent mixing in the ocean exhibits a near log-normal frequency distribution and elevated mixing events occur infrequently. However, these elevated mixing events are dominantly responsible for the vertical turbulent fluxes of solutes in the ocean. The 60+ profiles (I am guessing here as no numbers are provided in the manuscript) that may represent turbulence conditions over a period of 3 to 4 hours at the 15 to 20 individual stations are certainly inadequate to draw any conclusions on average turbulence quantities at different latitudes. The variability of turbulent mixing is also reflected by (2) the individual estimates of vertical turbulent nutrient fluxes available from the limited individual stations along the transect. Fluxes vary by three orders of magnitude (Figures 7, 8, 9). Again, an analysis of their statistical uncertainty would show the ambiguity of any trend analysis. Finally, (3) I cannot approve the approach chosen here as a whole. Comparing the strength of upper thermocline mixing at different latitudes cannot lead to any conclusions on local changes of the strength of turbulent mixing e.g. due to locally increasing stratification. The regions where measurements were taken combine very different external forcing and internal wave environments making it impossible to relate mixing strength to a single parameter.
*>>>In reply to point (1) We are aware of the near-lognormal PDF of turbulence dissipation data, actually it is one of the reasons to plot our data in log-fashion. We do not agree that the number of profiles cannot say anything about average quantities, as the spread is clearly given. Of course one can*

*compute average values from that, also considering that every 24 Hz sampled profile is binned in 7 m vertical bins (200 data points), that are again grouped in several layers (down to 500 m, or 70 bins). We wonder if the reviewer hereby discards all observational oceanographic turbulence work? Much effort goes into such observational work. Point (2) Yes, that is precisely what we indicated in the original manuscript: two (to four) orders of magnitude variability. The statistics is thereby given: the spread around the mean, considering the instrumental and methodological error of about half an order of magnitude. Point (3) We do not agree with this statement, because all sampling is done in the upper 500 m where the local water depth was at least 1100 m, and, except for 3 stations, most stations were over (much) deeper waters >2000 m. So, sampling was well away from bottom topography, in the NE-Atlantic where semidiurnal tides, and inertial motions, dominate the internal wave field, in summertime under overall moderate-good weather conditions across the entire survey. As a result, the dominant convection (in the upper 20-30 m) and internal wave induced mixing (in the stratified layers below) are much less variable across the transect due to different forcing than due to the highly intermittent occurrence of turbulent bursts as the reviewer correctly indicates above. Those bursts are inherent to turbulence, and less so dependent on the generation process. We added text to better explain this, lines 419-421:' If shear-induced turbulence in the upper ocean is dominant it may thus be latitudinally independent (Jurado et al., 2012; deeper observations present study). There are no indications that the overall open ocean internal wave field and (sub)mesoscale activities are energetically much different across the mid-latitudes.'*

As a revision strategy, I would advise the authors to remove the discussion on mixing and nutrient fluxes in a changing climate from the manuscript. Instead, the focus could be shifted to a detailed discussion of an upper-ocean nutrient budget including statistical uncertainties and a comparison to the net community production.
*>>> The outcome of our paper is the suggestion that climate change might not affect fluxes as strongly as current paradigm suggests. The intention is to inspire discussion/further research. The nutrient budget and comparison to the net community production have been described by Mojica et al (2016), which we will not repeat in our paper which is more oriented to physics processes than biology. We explained this better now. Our manuscript is an extension of that work.*

Some specific comments
Line 294 – 299, discussion of nutrient fluxes in the mixed layer and Fig. 7. I find the discussion of macro-nutrient fluxes in the mixed layer erroneous. First of all, vertical gradients of macro-nutrients are mostly insignificant. Macro-nutrient concentrations determined by a QuAAtro autoanalyser usually have accuracies of 0.1mM if CRM standards were used (please add details of uncertainties inherent to the nutrient concentrations to the methods section). To me, the differences between macro-nutrient concentrations measured at 10m and 25m depth are mostly smaller than measurement uncertainties.
*>>>The reviewer is right, we should have given the precision and detection limits. Without that info, the interpretation is not well substantiated. However, the accuracy is much better than assumed by the reviewer, as it is e.g. 0.028 µM for phosphate. We added the information now in the Methods section of the revised manuscript.*

*Absolute and relative precision for reasonably high concentrations in an in-house standard that is often measured.*

|        | S.D. (µM) | N   | concentration | rel SD |
|--------|-----------|-----|---------------|--------|
| PO4    | 0.028     | 30  | 0.9           | 3.1%   |
| NO3    | 0.143     | 30  | 14.0          | 1.0 %  |
| Si     | 0.088     | 15  | 20.99         | 0.42%  |

*The method detection limit was calculated during the cruise using the standard deviation of ten samples containing 2% of the highest standard used for the calibration curve and multiplied with the student's value for n=10, thus being 2.82. (M.D.L = Std Dev of 10 samples x 2.82)*

|         | *μM*   |
|---------|--------|
| PO4     | 0.007  |
| NH4     | 0.010  |
| NO3+NO2 | 0.012  |
| NO2     | 0.003  |
| Si      | 0.008  |

Line 300 – 303, discussion of nutrient fluxes below the mixed layer. As stated in the above, individual estimates of nutrient fluxes vary by three orders of magnitude and a statement about how the nutrient fluxes vary with latitude (i.e. with stratification) is inadequate. What may be interesting to the reader is the magnitude of average regional fluxes that could be compared to previous estimates (see e.g. Cyr et al., 2015). Presented results should also include nitrate/nitrite fluxes as the relative vertical turbulent fluxes of reactive nitrogen species and phosphorous could be of interest to a broader scientific community.

*>>>We are happy to compare with works from others, noting that Cyr et al. presented work at 2 stations in an estuary, which may be difficult to compare with the open ocean. We choose to graphical display macronutrient phosphorous representing other nutrients that show similar latitudinal trends. Attached is a version of Fig. 9 demonstrating the little extra information if we conclude NOx. We have to rescale panel c. We have now given global figures for nitrate fluxes. As mentioned, in this paper we are mainly interested in latitudinal and stratification trends and trends for phosphate fluxes precisely represent those for nitrate fluxes (blue and green lines in panel d in the figure below).*

[Figure]

Literature

Cyr, F., D. Bourgault, P. S. Galbraith, and M. Gosselin (2015), Turbulent nitrate fluxes inthe Lower St. Lawrence Estuary, Canada, J. Geophys. Res. Oceans, 120, 2308–2330,doi:10.1002/2014JC010272.

DeCarlo, T. M., K. B. Karnauskas, K. A. Davis,and G.T.F. Wong (2015), Climate modulates internal wave activity in the Northern South China Sea, Geophys. Res.Lett., 42, 831–838, doi:10.1002/2014GL062522.

---

## Author Comment (AC3) · 27 Oct 2020

*>>>We are grateful for the comments on our manuscript from the reviewer. We feel that this new version of the paper is much stronger as the result of the comments we received on the original manuscript. We have addressed all of the comments and have detailed our response to specific comments below. Our response to each comment is bulleted and in italics below the relevant comment behind>>>*

**Anonymous Referee #3**

This is an observational study of diapycnal mixing and the corresponding nutrient flux in the upper ocean across a quasi-latitudinal transect in the Northeast Atlantic Ocean. The data cover a rather long distance from 30deg.N to 62deg.N. The measurements were mainly temperature and conductivity profiles (from which the density or potential density profiles were obtained) with a carefully modified CTD system, and the turbulent kinetic energy dissipation rate and the diapycnal diffusivity were estimated based on the overturning (Thorpe) scale analysis. In general, the methodology of the analysis is reasonable, and useful information on turbulent mixing characteristics along the transect is obtained. However, I cannot recommend this manuscript for publication in the present form due to the major concerns as detailed in the following.
*>>>Thank you for the appreciation of the outline and methodology.*

First of all, I find the major point that the authors try to make (i.e., "nutrient availability for phytoplankton in the euphotic surface waters may not be affected by the physical process of global warming") is not convincing at all. For me, the point is not even relevant to what the data have shown. Obviously, the exact response of the upper ocean to global warming could be rather complicated, and I do agree with the authors that the global warming may not necessarily lead to a change in vertical turbulent exchange, but the results presented in the manuscript are by no means evidence for this. One may expect to see clear trend of upper ocean mixing (and corresponding material fluxes) at a certain location under continuing warming, but in such a large region covering more than 30 degrees, the underlying dynamics controlling diapycnal mixing could be very different from place to place, thus the spatial difference in mixing seen along the transect cannot be simply taken as a result of the difference in stratification (or "warming" by solar radiation).
*>>>There is general consensus that upper ocean convection and interior ocean internal wave breaking are the dominant turbulence generating mechanisms in the upper 50 – 100 m and in the deeper 100(50)-500 m stratified waters, respectively. These mechanisms are universal and not particularly location dependent, but do depend on variations in stratification. Turbulence is such an intermittent process that variations over four orders magnitude occur, at the same location (e.g. Gregg, JGR1989). Such variability is found in the present observations too, e.g. as indicated in old l.242 and 243. This is much larger variability than particular variation in turbulence generation processes in the ocean interior, away from boundaries. In other words, the sources may not greatly vary their energy content along a transect compared with turbulence intermittency. The trend in spatial difference in mixing along a transect can be taken as a result of difference in stratification. Along these lines we have added text (l 233-235) 'As will be demonstrated below, this is considerably less spread in values than the natural turbulence values variability over typically four orders of magnitude at a given position and depth in the ocean (e.g., Gregg, 1989).'*

More technically, although I do appreciate the authors' efforts in estimating turbulence and mixing characteristics from carefully conducted CTD measurements (via overturning scale analysis), I cannot be convinced by the subtle mixing (and flux) variability revealed by their estimates. As well acknowledged by the authors, even with the microstructure measurements one cannot expect to get an estimate with insignificant uncertainty. I agree that the overturning (Thorpe) scale analysis could be very useful in getting a rough estimate of mixing intensity when more direct measurements are not available, but using it to reveal subtle spatial (or temporal) variability could be misleading. For this purpose, direct

microstructure measurements are certainly much more reliable. On the other hand, ocean turbulence is certainly a stochastic process with both significant dynamical variability (which could be taken as deterministic linked to certain dynamical processes generating turbulence) and intermittency. As such, for the purpose of evaluating spatial variability of turbulent mixing, one should look at turbulence statistics. How many data points are used to get the reported averages? How does the PDF in each corresponding depth range look like? What are the confidence intervals of the reported averages? Are the noted differences/variabilities really significant?

*>>>We thank the reviewer for the appreciation of our efforts in estimating turbulence from carefully conducting CTD measurements. We like to point out that our near-surface data do show the same trends as upper 100 m microstructure profiler observations obtained a few years earlier along the same transect (Jurado et al 2012). In general, microstructure measurements are indeed more reliable for turbulence measurements, but in this case do not provide significantly different results. We note that turbulence is not by any means a deterministic process, even though its dominant generator, e.g. tide, may be deterministic. We have now additional CTD sampling numbers. The CTD sampled at 24 Hz, so 7 m ensemble averaging vertical intervals contain about 200 data points and we collected 3 to 6 CTD-casts per station. Below we add PDF of the entire dissipation rate data, averaged per vertical interval as indicated. As for the errors: they are about a factor of 3 for mean dissipation rate.*

[Figure]

*(New)Fig. A2. Probability Density Functions of logarithm of vertically averaged dissipation rate in comparison with latitudinal trend extreme values. (a) Distribution as a function of latitude for all data. (b) As a, but for the upper 15 m averages only. The mean value is given by the vertical purple line, with the horizontal line indicating +/- 1 standard deviation. The vertical light-blue lines indicate the best-fit value of the trend for 30° and 63°N. (c) As b, but for averages between -100 < z < -25 m. (d) As c, but for averages between -500 < z < -100 m.*

To conclude, I agree that the reported analysis gives useful information about mixing characteristics along the sampled transect, but without clear information of the underlying mechanisms and robust constraint on the reliability and significance of the reported mixing variability, one cannot be led to the points that the authors try to make. In particular, the results presented in the manuscript do not seem to lend any support to the authors' argument on the global warming impact on upper ocean mixing and nutrient flux trend. The authors may choose to simply emphasize their mixing estimates from the overturning scale analysis, with clear indication of the underlying uncertainties.

*>>>We adapted part of the discussion to better relate to comments by the reviewers, including toning down the relation with climate change, and we hope that as such we now made our point more convincing..*

---

## Author Comment (AC2)

*>>>We are grateful for the comments on our manuscript from the reviewer. We feel that this new version of the paper is much stronger as the result of the comments we received on the original manuscript. We have addressed all of the comments and have detailed our response to specific comments below. Our response to each comment is bulleted and in italics below the relevant comment behind>>>*

**Anonymous Referee #2**

The manuscript titled "Diapycnal mixing across the photic zone of the NE-Atlantic" by Haren et al. quantified the upper ocean nutrient flux using a custom modified CTD and nutrient measurements at discrete depths from a latitudinal transect along 17_5_W between 30 and 62_N in summer. The authors observed no increase in vertical mixing or diapycnal nutrient flux from south to north, where the temperature increased. Further, they opined that nutrient supply by diapycnal flux to the euphotic zone might not be affected by the physical process of global warming. It is a well-written manuscript and presents an interesting take on the ocean biophysical coupling in the global warming scenario. However, I feel that the authors jumped into a conclusion without providing enough evidence to support their say. Hence I recommend major revision.

*>>>Thank you for the appreciation. To be noted, temperature decreased, not increased, from south to north and we like to add that stratification, the medium to support internal gravity waves, also decreased.*

Major Comments

L63-96 The introduction needs a more general introduction to the oceanography of the region. Especially knowledge of bathymetry, background internal wave field, eddy kinetic energy, and wind conditions during summer.

*>>>We have no objection to add information on the North-Atlantic Ocean in general. However, the observations were made in the upper 500 m, and water at all stations were >1000 m deep with only 3 <2000 m. Local bottom topography did not influence the internal wave field directly. We added this consideration now in the revised manuscript. We also added that the survey was done in summer time, with in general moderate to good weather conditions, no big storms. We have no information on the eddy kinetic energy at the time, other than the generally excepted view from literature, which we also added to the revised manuscript.*

L123 In the Thorpe length calculation section, please mention the lowering speed of the CTD. A slow lowering can resolve overturns efficiently. In the mixed layer, the Thorpe method will consider it as a large overturn. How you will justify the validity of diffusivity within the mixed layer, where N2 is weak. A brief discussion on lowering speed of CTD and justification for the diffusivity within the mixed layer will give clarity to the reader.

*>>> We agree with the reviewer that we should have added the lowering speed, it was 1 m $^{-1}$s. Yes, slow lowering resolves overturns better, but in doing so it is lowered obliquely through the overturns in case of non-zero background flow, which is nearly always present. A completely and thoroughly mixed layer hardly ever exists, but the stratification is often weak in the upper 20-30 m while is varies in height and time. For the validity of choice of parameters we refer to the extensive work by Oakey (1982) who demonstrated upper ocean parametrization to vary over at least one order of magnitude but, given enough data points, with a particular average value as used here. This is confirmed in more recent works (Gregg et al 2018) for ocean observations and Portwood et al (PRL2019) for modelling work. We added this information to the manuscript at P14: 'Although the general understanding, mainly amongst modellers, is that the Thorpe length method overestimates diffusivity (e.g., Scotti, 2015; Mater and Venayagamoorthy, 2015), this view is not shared amongst ocean observers (e.g., Gregg et al., 2018). In the large parameter space of the high Reynolds number environment of the ocean, turbulence properties vary constantly, with an interminglement of convection and shear-induced turbulence at various levels. Given sufficient*

*averaging, and adequate mean value parametrization, the Thorpe length method is not observed to overestimate diffusivity. This property of adequate and sufficient averaging yields similar mean parameter values in recent modelling results estimating a mixing coefficient near the classical bound of 0.2 in stationary flows for a wide range of conditions (Portwood et al., 2019). It is noted that diffusivity always requires knowledge of stratification to obtain a turbulent flux, and it is better to consider turbulence dissipation rate for intercomparison purposes. Nevertheless, future research may perform a more extensive comparison between Thorpe scale analysis data and deeper microstructure profiler data.'*

L256-258 Substantiate the surface cooling and internal wave breaking using data.
*>>>This was indeed not clear. We meant that the main process in the upper layer is convective surface cooling, and internal wave breaking in the more stratified layers below. We changed the text l.289-291: The trends suggest only marginally larger turbulence going poleward, which is possibly due to larger cooling from above and larger internal wave breaking deeper down.*

L264-265 I could not understand this sentence.
*>>>Perhaps 'confirm' was misused here; we meant to say that the deeper layers show the same latitudinal trend in turbulence and stratification values as the upper layer. We rephrased the sentence to' The data from well-stratified waters deeper down thus show the same latitudinal trend as the observations from the near-surface layers.' (line 298-300).*

L284-286 The nutrient flux depends on the eddy diffusivity and the nutrient concentration gradient, which changes dramatically with depth. The nutrient fluxes thus may vary with two-or-three orders difference. In the manuscript, nutrient flux is calculated using a low-resolution profile of nutrients. Does this discrete measurement introduce bias to the flux calculation? What is the typical depth of the euphotic zone in the study region?
*>>>We would have liked a denser sampling of nutrients, but that was impossible in the cruise plan. On the other hand, the large gradients in nutrients are indeed in the vertical, and variations in the horizontal plane are less strong. We note that, due to overturn sizes over which we must average, turbulence is gridded in equally large vertical distances. The typical depth of the 0.1% irradiance penetration is about 50 to 100 m, see the figure panel 'a' below in which we compare this depth with the 'mixed layer depth', defined as the depth at which the temperature difference with respect to the surface was 0.5°C (as in Jurado et al 2012). We have added this information on p.12: 'The mixed layer depth, defined as the depth at which the temperature difference with respect to the surface was 0.5□C (Jurado et al., 2012), varies between about 20 and 30 m on the southern end of the transect and weakly becomes shallower with latitude (Fig. 7a). This weak trend may be expected from the summertime wind conditions that also barely vary with latitude (Fig. 7b,c). In contrast, the euphotic zone, defined as the depth of the 0.1% irradiance penetration level (Mojica et al., 2015), demonstrates a clear latitudinal trend decreasing from about 150 to 50 m (Fig. 7a).'*

[Figure]

*(New) Figure 7. Latitudinal transect of near-surface layers and wind conditions measured at stations during the observational survey. (a) Mixed layer depth (x) and euphotic zone (o). (b) Wind speed. (c) Wind direction.*

L318-320 General understanding is that the Thorpe length method overestimates the diffusivity (Mater and Venayagamoorthy 2015; Alberto Scotti 2015).

>>>*That is indeed a general understanding amongst modellers, but not amongst ocean observers (e.g. Gregg et al 2018). In the high Reynolds number environment of the ocean turbulence properties vary constantly, an interminglement of convection and shear-induced turbulence at various levels. Given sufficient averaging, and adequate mean value parametrization, the Thorpe length method does not overestimate diffusivity, see also recent modelling results by Portwood et al (PRL2019). It is noted that diffusivity always requires knowledge of stratification to obtain a turbulent flux, and it is better to consider turbulence dissipation rate for intercomparison. We clarified this in the revised manuscript (lines 364-377): 'Although the general understanding, mainly amongst modellers, is that the Thorpe length method overestimates diffusivity (e.g., Scotti, 2015; Mater and Venayagamoorthy, 2015), this view is not shared amongst ocean observers (e.g., Gregg et al., 2018). In the large parameter space of the high Reynolds number environment of the ocean, turbulence properties vary constantly, with an interminglement of convection and shear-induced turbulence at various levels. Given sufficient averaging, and adequate mean value parametrization, the Thorpe length method is not observed to overestimate diffusivity. This property of adequate and sufficient averaging yields similar mean parameter values in recent modelling results estimating a mixing coefficient near the classical bound of 0.2 in stationary flows for a wide range of conditions (Portwood et al., 2019). It is noted that diffusivity always requires knowledge of stratification to obtain a turbulent flux, and it is better to consider turbulence dissipation rate for intercomparison purposes. Nevertheless, future research may perform a more extensive comparison between Thorpe scale analysis data and deeper microstructure profiler data'.*

L328-329 Here you can add a detailed discussion on how internal waves can be a feedback mechanism to counteract the suppression of mixing by increased stratification.

*>>> Although originally it was merely meant as an introductory sentence to the paragraphs below, we see reviewer's point and we added it in the revised manuscript. (line 384-386 and pages that follow): 'We hypothesize that internal waves may drive the feed-back mechanism, participating in the subtle balance between destabilizing shear and stable (re)stratification as outlined below.'.*

L344-364 Authors need to provide data evidence to prove that Internal wave energy/eddy kinetic energy is more in Northern stations, and thus, the relatively increased stratification (compared to south) could not suppress the diapycnal flux of nutrients to the euphotic zone from deeper depths. This will give the readers a better understanding of the lack of correspondence between temperature /stratification and diapycnal flux with latitude.

*>>>There seems to be a misunderstanding here: Stratification is less in the north, compared to the south. We have emphasized this in the text. We would have loved to include direct observational information on internal wave and eddy kinetic energy but we do not have such data available in the present study. Instead, we refer to previous work in which we had such data. Using that information, we now better tried to explain as the reviewer suggests. In the discussion we support our suggested hypothesis with the (previous) observation that the state of ocean is one of marginal stability, in which stratification is a subtle balance between internal wave shear and -breaking.*

One could employ the GM spectrum calculated using gridded Historical data sets (ARGO) to give an idea on the background Internal wave energy. However, I won't insist on doing this analysis.

*>>>We think it is better that modelers take up this task, they will perform much better than we can on this.*

A discussion on the meteorological conditions during the observation period is also warranted. What if the southern stations were characterized with anomalously calm weather that mixing was inactive and became comparable to the northern stations.

*>>>This is a good idea, we added this information. For the information of the reviewer: meteorological conditions were moderate to good throughout the cruise, see for Wind Speed the panel b (and c for Wind direction) in the figure given above. This is the new Fig. 7 now.*

References
Scotti, A. (2015). Biases in Thorpe-scale estimates of turbulence dissipation. Part II: energetics arguments and turbulence simulations. Journal of Physical Oceanography, 45(10), 2522-2543.

Mater, B. D., Venayagamoorthy, S. K., St. Laurent, L., & Moum, J. N. (2015). Biases in Thorpe-scale estimates of turbulence dissipation. Part I: Assessments from largescale overturns in oceanographic data. Journal of Physical Oceanography, 45(10), 2497-2521.

---

## Author Response (AR2)

Dear editor,

We thank the reviewers for their constructive help in improving our manuscript and reviewers #2 and #3 for accepting our revised manuscript as is for publication.
Below you will find our latest replies to further queries by reviewer #1, in blue and italic.

Best regards,
Hans van Haren
(also on behalf of the coauthors)
* * *
Review (#1) for the reviesed manuscript # os-2020-73" Diapycnal mixing across the photic zone of the NE-Atlantic" by van Haren et al.

As indicated in the previous review of this manuscript, this paper discusses dissipation rates of turbulent kinetic energy, eddy diffusivities and vertical turbulent nutrient fluxes inferred from upper-ocean hydrographic and nutrient data taken during a cruise on a transect from 60°N to 30°N along about 17°W in the North Atlantic. While the new version of the manuscript is somewhat improved, in particular by adding statistical and measurement uncertainty, I still find that results presented in the manuscript are not sufficient to support the authors' interpretations and conclusions.

Remaining major concerns.
My major remaining concern continues to be the authors' claim "the lack of correspondence between turbulent mixing and stratification along the transect suggests that nutrient availability for phytoplankton in the euphotic surface waters may not be affected by global warming", which is not supported by their results. As stated in my first review, comparing the strength of thermocline mixing in different oceanic regions (at different latitudes, here) cannot lead to any conclusions on local changes of the strength of turbulent mixing due to locally increasing stratification. As I am detailing in the response below, the added wording does not help to justify their claim.
Furthermore, my second concern related to the presented vertical nutrient fluxes due to turbulent mixing has not fully been addressed. Although adding accuracy and precision of the nutrient measurements has improved the manuscript, it remains unclear if the presented nutrient fluxes are relevant for near-surface primary production and thus for the biological carbon pump as a whole. In my earlier review, I suggested to the authors to compare the results to previous studies and to provide a comparison to nutrient uptake in the upper ocean. Unfortunately, it seems that my suggestion was misinterpreted by the authors.

*We disagree with lack of proof for our general conclusion, which is now further weakened and elaborated in its wording. As will follow in responses in more detail below, the objections given by the reviewer are either not relevant or inaccurate. We acknowledge that we should have been more clear on these two points (internal wave characteristics and nutrient fluxes), as we do now, below and in the manuscript. In response to the reviewer's last sentence above: No, it was wrongly indicated, as the reviewer admits below.*

I provide further details to these concerns in my reply to the responses below. For clarity, I am presenting my former remarks, then the reply by the reviewers followed by my reply.

Reviewer's previous major remarks: (3) I cannot approve the approach chosen here as a whole. Comparing the strength of upper thermocline mixing at different latitudes cannot lead to any conclusions on local changes of the strength of turbulent mixing e.g. due to locally increasing stratification.

Reply by the authors: We do not agree with this statement, because all sampling is done in the upper 500 m where the local water depth was at least 1100 m, and, except for 3 stations, most stations were over (much) deeper waters >2000 m. So, sampling was well away from bottom topography, in the NE-Atlantic where semidiurnal tides, and inertial motions, dominate the internal wave field, in summertime under overall moderate-good weather conditions across the entire survey. As a result, the dominant convection (in the upper 20-30 m) and internal wave induced mixing (in the stratified layers below) are much less

variable across the transect due to different forcing than due to the highly intermittent occurrence of turbulent bursts as the reviewer correctly indicates above. Those bursts are inherent to turbulence, and less so dependent on the generation process. We added text to better explain this, lines 419-421:' If shear-induced turbulence in the upper ocean is dominant it may thus be latitudinally independent (Jurado et al., 2012; deeper observations present study). There are no indications that the overall open ocean internal wave field and (sub)mesoscale activities are energetically much different across the mid-latitudes.

Reply by the reviewer: I do not want to get into a (perhaps endless) discussion on internal wave forcing and energy fluxes. However, even if the energy residing in the internal wave field were comparable at the different latitudes (which the authors do not demonstrate in this study nor was this shown in the study by Jurado et al. 2012) it can not be concluded that the energy fluxes into internal waves and from internal waves into turbulence would be the same. In fact, this would be rather unlikely, because internal wave-wave interaction processes are dependent on latitude (see e.g. Henyey et al. 1986 or Gregg et al. 2003). Wave-wave interaction at higher latitudes is more efficient in fluxing energy to turbulence than at lower latitudes and we would thus expect an elevated flux of internal wave energy to turbulence at 60°N compared to 30°N. Furthermore, parametric subharmonic instability of the M2 tide leads to elevated energy flux into turbulence and enhanced mixing in the region between about 20°-30° away from the equator (see e.g. Hibiya et al. 2007). This process may be responsible for the somewhat elevated eddy diffusivities in the southern part (30°N-32°N) of the transect. I am mentioning these two processes because the added text, in particular "no indications that the open-ocean internal waves field … are energetically not much different across the mid-latitudes" does not help to understand the claim that eddy diffusivities across the transect are of comparable magnitude.
Apart from issues with the added sentence, I strongly disagree with the statement in the authors response above "… internal wave induced mixing (in the stratified layers below) are [is] much less variable across the transect due to different forcing than due to the highly intermittent occurrence of turbulent bursts …" In the thermocline, internal wave energy is dissipated by turbulence. Certainly, the bursts of turbulence are highly variable, but if resolved adequately in time, they merely reflect the dissipation of internal wave energy. There are numerous factors that impact local internal wave energetics and their energy flux to turbulence (see e.g. McKinnon et al. 2017 for a recent review). Additionally, wind stress curl may efficiently excite near-inertial waves if rotating anticyclonically at frequency close to the Coriolis frequency even at low wind speeds.
I retain my position that the authors should remove the discussion on mixing and nutrient fluxes in a changing climate from the manuscript. The data analysis presented in the study does not allow to draw conclusions on this matter.

*There seems to be a misunderstanding by the reviewer. Previous observations (van Haren, 2005b, and Hibiya et al., 2007) have shown that a diurnal critical latitude enhancement of near-inertial internal waves only occurs sharply equatorward of |30°| (not 32°) latitude. The present observations are all made poleward of this latitude. Likewise, earlier observations (van Haren 2005a) have demonstrated that the Henyey/Gregg model does not hold close to the equator, where the internal wave regime drops much more rapidly in a non-gravity wave system. The H/G model is not varying very much across mid-latitudes, maximum by a factor of 1.8 between 30° and 63°, compared with the observed variations and errors (factor of 3) in turbulence dissipation rate. Naturally, other processes like interaction between internal waves and mesoscale phenomena may be important, but these occur in a similar fashion across the sampled ocean far away from boundaries. As mentioned in our revised manuscript, all observations were made (i) under similar summertime weather conditions, providing comparablenear-inertial internal wave generation, and (ii) far away from major topographic features, i.e. r from internal tide sources, (which, if important, would have definitely given different results at our station at the latitude of Porcupine Bank, for example). The above is claried even better now in the manuscript.*

Reviewer's previous comment: As a revision strategy, … Instead, the focus could be shifted to a detailed discussion of an upper-ocean nutrient budget including statistical uncertainties and a comparison to the net community production.

Reply by the authors: The outcome of our paper is the suggestion that climate change might not affect fluxes as strongly as current paradigm suggests. The intention is to inspire discussion/further research.

The nutrient budget and comparison to the net community production have been described by Mojica et al. (2016), which we will not repeat in our paper which is more oriented to physics processes than biology. We explained this better now. Our manuscript is an extension of that work.

Reply by the reviewer: I have two issues with the revised version and the response above. First of all, my last suggestion to refer to the publication by Cyr et al. (2015) was misunderstood, because I did not want to point to the results of that particular study but rather their table 1 listing about 20 published studies on vertical nutrient fluxes by turbulent mixing on page 2326. I should have been more detailed with my comment, here. This list also includes several studies in the subtropical and subpolar North Atlantic. In particular, the results by Martin et al. (2010) are of relevance here. Their measurement program was conducted in boreal summer in an area (PAP, 49°N 16°30'W) very close to the transect which data are analyzed here. The study by Martin et al. revealed vertical nutrient fluxes due to turbulent mixing that are very similar to the numbers reported by the authors here. However, additional analysis by Martin et al. showed that these nutrient fluxes account for only about 2% of the nutrient uptake within the euphotic zone. They conclude that other processes must be responsible to supply nitrate to the euphotic zone that are much more relevant than vertical fluxes due to mixing.

In the study by Mojica et al. (2016), no numbers for the calculated vertical turbulent nutrient fluxes are given. This is different from the study here. When comparing these numbers with previous studies, it appears that they are too small to sustain significant production in the euphotic zone. Is it possible that rare intense mixing events relevant for the vertical supply of nutrients were not captured by the measurement program (see e.g. Hummels et al., 2020)? A discussion of this issue needs to be added to the manuscript.

*In reply to the reviewer's comment on rare mixing events:*
*If important, rare mixing events would show up in the data, for example in data by Martin et al. (2010) who stayed in the same station for a prolonged period. At least they would see differences in day/night turbulence, whereby we note that convection is also a vertical turbulent exchange process. As mentioned previously, ocean mixing is characterized by high intermittency, of a puff here and a puff there, which causes single dissipation rates to vary over at least four orders of magnitude. The same spread in values characterizes 'rare events' as observed by Hummels et al. (2020) in the equatorial Atlantic, the same area studied by Alford and Gregg (2001), an area well outside our transect.*

*Concerning the vertical nutrient fluxes comment by the reviewer:*
*Our cruise transect entails a latitudinal gradient from 30 to 63 degrees N, which also includes the latitude at which the study by Martin et al. was conducted. It is promising that our vertical nutrient fluxes indeed match those obtained in the study by Martin et al. (2010), which gives confidence in the methods applied. The comment by the reviewer that Martin et al. (2010) state that such nutrient flux only accounts for 2% of the uptake within the euphotic zone can however not be generalized. The nitrate uptake assays by Martin et al. (2010) spiked 100-200 µL stock solution $K^{15}NO_3^-$ of 1 µmol/100 µL concentration to 2 L sample water in order to obtain approximately 10% of the ambient dissolved nitrate concentration. As such, a minimum spike of 100 µl per 2 L sample implies there was a minimum of 5 µmol/L nitrate at their location. Given the calculated uptake of nitrate of 0.1 µmol/L/day, this implies (i) the population could be sustained by about 50 days on the existing nutrient inventory without any additional flux, and (ii) that most likely something else must have been limiting the phytoplankton growth (as otherwise the inventory would be depleted much faster). Inventories along our transect were much lower and the two studies are thus not directly comparable. We suspect that in the case of Martin et al. (2010), either a past upwelling event or lateral advection (their study site is close to the European continental shelf) led to the enhanced inventories, and consequently higher nitrate uptake rates were possible. Thus, while the vertical turbulent fluxes are similar, it is unlikely the uptake rates should be similar.*
*Furthermore, particularly in the southern part of our cruise transect remineralization of N in the form of ammonia likely supplied most of the N-demand. A small flux of nitrate from below may be sufficient to balance losses of N via export to the deeper waters. For example, Gaul et al. (1999), found that nitrogen regeneration could dominate nitrogen supply and recycling efficiencies up to 100% under post-bloom situations. Our earlier work largely along the same transect (Mojica et al. 2016) demonstrated (i) that phytoplankton cell losses generally matched the production, and (ii) that viral lysis was equally responsible for those losses as grazing. Viral lysis has been shown to release labile organic matter, strongly stimulating bacterial recycling of nutrients (ammonia and phosphate), and at the same time zooplankton are known to contribute substantially to ammonium release.*

*In our opinion, the relatively low turbulent flux in the stratified waters of our summer cruise aids to a low F-ratio, rather than an important nitrate flux is missing. That being said, supply from above via aeolian deposition as well as nitrogen fixation by diazotrophs can play an important role, but this is beyond the scope of this paper that focusses on differences (or lack thereof) in observed turbulent fluxes along a meridional transect.*

References

Alford, M. H. and Gregg, M. C.: Near-inertial mixing: Modulation of shear, strain and microstructure at low latitude, J. Geophys. Res., 106, 16,947-16,968, 2001.

Cyr, F., D. Bourgault, P. S. Galbraith, and M. Gosselin (2015), Turbulent nitrate fluxes in the Lower St. Lawrence Estuary, Canada, J. Geophys. Res. Oceans, 120, 2308–2330.

Gaul, M., A.N. Antia, W. Koeve (2015), Microzooplankton grazing and nitrogen supply of phytoplankton growth in the temperate and subtropical northeast Atlantic, Mar. Ecol. Progr. Ser., 189, 93-104.

Gregg, M. C., T. B. Sanford, and D. P. Winkel (2003), Reduced mixing from the breaking of internal waves in equatorial waters, Nature, 422, 513–515.

Henyey, F. S., J. Wright, and S. M. Flatte (1986), Energy and action flow through the internal wave field - an eikonal approach, J. Geophys. Res., 91(C7), 8487-8495.

Hibiya T., M. Nagasawa and Y. Niwa (2007) Latitudinal dependence of diapycnal diffusivity in the thermocline observed using a microstructure profiler, Geophys. Res. Lett, 34, L24602.

Hummels, R., M. Dengler, W. Rath, G. Foltz, F. Schütte, T. Fischer and P. Brandt (2020), Surface cooling caused by rare but intense near-inertial wave induced mixing in the tropical Atlantic, Nature Comm., 11, 3829.

Martin, A. P., M. I. Lucas, S. C. Painter, R. Pidcock, H. Prandke, H. Prandke, and M. C. Stinchcombe (2010), The supply of nutrients due to vertical turbulent mixing: A study at the Porcupine Abyssal Plain study site in the northeast Atlantic, Deep Sea Res., Part II, 57(15), 1293–1302.

MacKinnon, J. A., et al. (2017) Climate Process Team on Internal Wave–Driven Ocean Mixing. Bulletin of the American Meteorological Society 98, 2429-2454.

Mojica, K. D. A., Huisman, J., Wilhelm, S. W. and Brussaard, C. P. D.: Latitudinal variation in virus-induced mortality of phytoplankton across the North Atlantic Ocean, ISME J., 10, 500-513, 2016.

van Haren, H. 2005a. Sharp near-equatorial transitions in inertial motions and deep-ocean step-formation. Geophys. Res. Lett., 32, L01605, doi:10.1029/2004GL021630.

van Haren, H., 2005b. Tidal and near-inertial peak variations around the diurnal critical latitude. Geophys. Res. Lett., 32, L23611, doi:10.1029/2005GL024160.

---

## Author Response (AR3)

Dear Editor, Dear Ilker,

The best wishes for 2021 to you too!
We much appreciate your critical comments. You will find ou replies below, and a 'track changes' version with the items and text inserted.

Also on behalf of my co-authors,

Best regards,

Hans

Topic Editor Decision: Publish subject to minor revisions (review by editor) (01 Jan 2021) by Ilker Fer
Comments to the Author:

Dear Hans,

Happy 2021! Thanks for the revised submission of your manuscript.

I think you have addressed the comments of the reviewers satisfactorily and I do not see the need to send the manuscript out for further comments. I doubt that you and reviewer 3 will converge; however, together with the open discussion and concerns raised, your findings and analysis can be documented in the literature.

I have some other concerns, which must be addressed before the manuscript could be published. I list them below together with some minor issues.

Thank you,

Ilker

1- Description of the Thorpe scale analysis (Section 2.2):

Using Eq(1) in the form of Thorpe 1977 but with the 0.64 factor from Dillon is misleading. In this equation, "d" must be replaced with the Thorpe scale, LT which is the r.m.s. of d over the overturn (or over constant vertical scale of 7 m as you calculated). Although not ideal, it is acceptable to use a fixed vertical averaging scale. Please revise throughout by introducing LT = rms(d), and in line 223, LO/LT = 0.8.
>>>*Done as you suggest, although it was meant to be introduced two paragraphs later, and we did prefer to use |d|_rms over L_T.*

Also because potential density is introduced for analysis (line 211) (and also C_T and S_A earlier), the exact definition of N using in situ density is confusing. Please start the description using sigma_theta from line 204 (and N approximated as sqrt(g/rho0 dsigma_theta/dz) ). Also, because rho changes in space, use partial derivative with z.
>>>*Yes, that is true, it came from an attempt to keep the paper readable also for non-physical oceanography colleagues. We have restructured largely following your suggestions. So, we now use potential density from (old) l.204 and give the practical definition of N. But we retain the other, exact definition, and insert some text to warn people, following e.g. Gill (1982) and King et al. JGR2012, to use that definition for deeper waters. As buoyancy frequency is not a vector but a scalar, and although we agree it may vary over (x,y), we retain the total derivative following Gill (1982).*

2. Vertical scale for N (100 m) versus LT (7 m):

This leads to at least two issues. Firstly, the Thorpe scale analysis is typically based on a background stratification over the overturn scale. There is a mismatch (and inconsistency) between 7 m and 100 m. Secondly, when mixed layer depth is about 30 m (Fig 7a), calculating N over 100 m will always give you unrealistically large stratification in the mixed layer. Your analysis (of epsilon, Krho and turbulent fluxes) in the upper layer (0-15 m averages) will be biased. All green data points in Fig 5 are likely in error. Furthermore, given that MLD is 30 m or so (i.e., > 15 m), by definition N2 = 0 (or at least cannot be resolved by density profiles), and the application of the Osborn approach (for Krho) in the upper 15 m is not allowed since N2 leads to singularity. One approach is to exclude the 0-15 m results from the paper.

*>>>The 7 m is chosen on the basis of the Ozmidov scale, as a compromise. The 100 m scale refers to N computed over raw potential density profiles, not the reordered profiles. N computed over the reordered profile using the 7-m scale mends this (yielding very much the same result as using 100-m scale over raw profiles), under (Thorpe method's) assumption that the difference between raw and reordered profile is entirely attributable to mechanical turbulent overturning.*

*The MLD is defined, following previous conventions, as the level where a nonzero threshold (DeltaT>0.5 degrC) is passed, so in practice N^2 is nonzero: If one carefully investigates the potential density profile, the 'MLD' is not (everywhere) fully mixed and using a criterion of 0.001 degrC of a reordered profile demonstrates an 'MLD' of only about 5 m. Recall that we use 7 m scales for computing N from reordered profiles.*

*As the paper is also aiming at a readership of marine biologists who work in the euphotic zone and as the results from 0-15 m are consistent with those deeper down, which indirectly confirms the improvement of the CTD-modification which is also supported by previous ner-surface microstructure observations by Jurado et al (2012), we prefer to leave them in. However, triggered by your comment, we now include more cautionary notes on these data.*

In any case I would recommend repeating the analysis using 7 m vertical scale for both N2 and LT. I would also recommend screening the data (excluding epsilon and Krho) over segments when there are very few displacements in a 7-m window, and when N is less than a noise level (hence Krho and turbulent fluxes undefined).

*>>>N (from the reordered profile) and LT were calculated over 7 m. The turbulence data are screened when they fall below threshold (cf x-axis limits and gaps in profiles in Figs 3,4 panels c,d).*

3. Turbulent fluxes (and Fig 8-9): I find the presentation of figures 8-9 confusing. Some data points are excluded without mentioning. Take the vertical gradient value, first red circle data point in panel c. It has no corresponding data in panel a. The next data points near 33N in panel a do not have a corresponding gradient in panel c but have a flux value in (d). Please carefully go through what you plot and ensure the dataset is correctly presented. At latitudes 60-63N blue dots and circles are almost collocated in panel b, hence zero vertical gradient, yet there are substantial gradient values (blue crosses) in panel c. This is all very confusing.

In Fig 9, panel a, the second and third set of data points have the opposite sign gradient (dot and circles are reversed). But their turbulent fluxes are plotted positive. This is erroneous. The authors appear to have ignored the sign of turbulent fluxes (also in other occasions. e.g. DFe at 53N and PO4 at 30N, these must have a gradient sign opposite to other data points.

*>>>Thank you very much for your critical eye, you struck some embarrassing mix-up of errors due to mismatching of data-files and Matlab's inadequacy of blocking imaginary data (logarithm of negative values). We have gone through it all carefully and mended the errors, whilst only plotting downgradient values.*

*Yes, we agree that in Fig. 8 only few gradient values represent upward fluxes and that those gradients are very weak within the standard deviations of the measurements. But, as stated above, we would like to retain these results from the euphotic zone also because they are consistent with those from deeper down. We put in extra cautionary words.*

*All in all, we replotted Figures 1,3,4,5,8,9 and 10*

Minor points:

Li52: delete "near the surface"
*>>>OK*

Li 54: [average] temperature decreased
>>> *'average' inserted now*

Li 111: end the paragraph with a concluding sentence that the sampled dataset is adequate for a discussion on the variability of turbulence, stratification and turbulent nutrient fluxes with latitude.
>>>*Concluding sentence inserted*

Li 175: [A]bsolute [S]alinity, S_A (that is capital first letters, and subscript A)… [potential] density anomalies…
>>>*Modified as suggested*

Li 231: should be Gregg et al (2018)
>>>*Yes, thank you*

Li 238: downgradient turbulent fluxes, with z traditionally defined upward has a minus sign, i.e. -Kz d()/dz.
>>>*Added 'downgradient' and minus-sign*

Li 246: Where does 7x10-5 come from? If it is half of peak-to-peak raw variations of 1.4x10-3, it should be 7x10-4? Note this value (7x10-5) is mentioned again in Fig 3's caption.
>>>*The 7x10^-5 follows from standard deviations determined across short sections of near-homogeneous layers in the potential density profile. This corresponds to noise-variational amplitudes of 1.4x10^-4 in raw data, see also the small-scale variations in (detrended) raw temperature data in Fig. 2b for an indication. Sorry for the small misinformation in the manuscript. The correct threshold is 7x10^-5 as indicated.*

Li 304: dissipation rate[s]
>>>*Yes, modified*

Fig 3: In practice a proper application of Thorpe scale analysis with detection of overturns (even with very high-quality density measurements) cannot resolve epsilon values of 10-11 or less, yet panel c shows such small values. It must be an artifact of using dz=7 m and that 7 m window having a few non-zero d values, resulting in a low epsilon. A data screening excluding segments with few data points can improve this.
>>>*Yes we agree, the minimum level is about 10^-11, so we blocked panels c in Figs 3and 4 at minimum level now. Data gaps also indicate values below threshold.*

Fig 3. Li 748: relaced average over with r.m.s calculated over
>>>*Yes, indeed*

Fig 3, 4 and throughout as needed: x-label for N should be log(N) as with other parameters. Also in the caption of Fig 3 (and other captions where log is mentioned) note that it is logarithm base 10.
>>>*Modified now; we follow the mathematics ISO 80000 recommendation to write log_10(x) as lg(x), now indicated in the caption of Fig. 3 and used throughout.*